# Gene Expression Changes in the Ventral Tegmental Area of Male Mice with Alternative Social Behavior Experience in Chronic Agonistic Interactions

**DOI:** 10.3390/ijms21186599

**Published:** 2020-09-09

**Authors:** Olga Redina, Vladimir Babenko, Dmitry Smagin, Irina Kovalenko, Anna Galyamina, Vadim Efimov, Natalia Kudryavtseva

**Affiliations:** 1FRC Institute of Cytology and Genetics, Siberian Branch of Russian Academy of Sciences, 630090 Novosibirsk, Russia; bob@bionet.nsc.ru (V.B.); smagin@bionet.nsc.ru (D.S.); koir@bionet.nsc.ru (I.K.); galyamina@bionet.nsc.ru (A.G.); efimov@bionet.nsc.ru (V.E.); n.n.kudryavtseva@gmail.com (N.K.); 2Department of Natural Sciences, Novosibirsk State University, 630090 Novosibirsk, Russia

**Keywords:** ventral tegmental area (VTA), daily agonistic interactions, RNA-Seq, chronically winning mice (winners), chronically defeated mice (losers)

## Abstract

Daily agonistic interactions of mice are an effective experimental approach to elucidate the molecular mechanisms underlying the excitation of the brain neurons and the formation of alternative social behavior patterns. An RNA-Seq analysis was used to compare the ventral tegmental area (VTA) transcriptome profiles for three groups of male C57BL/6J mice: winners, a group of chronically winning mice, losers, a group of chronically defeated mice, and controls. The data obtained show that both winners and defeated mice experience stress, which however, has a more drastic effect on defeated animals causing more significant changes in the levels of gene transcription. Four genes (*Nrgn*, *Ercc2*, *Otx2*, and *Six3*) changed their VTA expression profiles in opposite directions in winners and defeated mice. It was first shown that *Nrgn* (neurogranin) expression was highly correlated with the expression of the genes involved in dopamine synthesis and transport (*Th*, *Ddc*, *Slc6a3*, and *Drd2*) in the VTA of defeated mice but not in winners. The obtained network of 31 coregulated genes, encoding proteins associated with nervous system development (including 24 genes associated with the generation of neurons), may be potentially useful for studying their role in the VTA dopaminergic neurons maturation under the influence of social stress.

## 1. Introduction

The model of repeated agonistic interactions (model of chronic social conflicts) is known as an effective approach for studying the physiological and molecular mechanisms involved in the formation of distinctive behavior patterns in mice under aggressive social encounters [1,2,3,4]. Under experimental exposure to daily agonistic interactions, two groups of animals are formed: (1) with experience of social victories (winners) and (2) with chronic experience of defeats (defeated mice or losers). Mice with repeated winning experiences in everyday agonistic interactions develop marked aggressiveness, anxiety, impulsivity, and impaired motivated and cognitive behaviors [3]. Mice with chronic experience of social defeats develop mixed anxiety/depression-like states [5,6].

In previous publications of our group, it was shown that stress resulting from daily agonistic interactions causes crucial changes in the expression of many genes in various brain regions of adult mice [5,7,8,9,10,11,12,13]. Earlier, we described in detail changes in the transcription of the genes associated with the neurotransmitter systems functioning in the ventral tegmental area (VTA) of defeated male mice as compared to the controls that did not have an agonistic interaction experience [9].

The VTA is known as a brain region from which the mesocortical and mesolimbic dopaminergic pathways originate. More than half of the VTA neurons are dopaminergic, however, non-dopamine-releasing VTA neurons also contribute significantly to VTA functioning. Both dopamine-releasing and non-dopamine-releasing VTA neurons participate in the generation of output firing patterns by the integration of afferent signals with local inhibitory or excitatory inputs [14]. Ascending dopaminergic fibers of VTA neurons are projected onto multiple cortical and subcortical regions, mainly the nucleus accumbens and medial prefrontal cortex [15]. VTA neurons are thought to play a central role in reward circuits, attention, and motivational, emotional and addictive behaviors. It should be emphasized that due to functional and anatomical heterogeneity, VTA dopamine neurons are activated by opposite emotional stimuli, such as reward and aversion. The induction of the VTA dopaminergic activity is considered to play a critical role in the development of a number of psychiatric disorders under both acute and repeated stress [16,17].

The studying of the molecular mechanisms underlying the excitation of VTA neurons, in the context of the formation of alternative forms of social behavior under long-term negative and positive experiences in daily agonistic interactions, may help to identify the genes and the metabolic pathways contributing the most to the alternative behavioral phenotypes such as depression-like states and pathological aggression.

The current paper provides a comparative analysis of differentially expressed VTA genes (DEGs) in winners and defeated animals (losers) compared with controls in order to identify the molecular determinants underlying the formation of alternative behavior patterns in male mice under daily agonistic interactions.

## 2. Results

### 2.1. Differentially Expressed Genes (DEGs) in the VTA of Winners vs. Control Mice

A total of 13,750 expressed genes were identified. Among them, 44 genes were characterized as being differentially expressed in the VTA of aggressive mice (winners) compared to controls (Appendix A
Appendix A). Of these, 28 genes (63.6%) had a lower level of transcription in the winners compared to controls. A functional annotation of DEGs revealed 14 genes associated with a behavior and neurological phenotype (Table 1). One of these genes (*Tph2*) is known as associated with abnormal aggression-related behavior and increased aggression towards mice. Three genes from the list (*Slc17a7*, *Slc6a4* and *Tph2*) are associated with the terms abnormal fear/anxiety-related behavior and abnormal depression-related behavior. Increased expression levels of transcription of the *Tph2* and *Slc6a4* genes suggest an increase in the synthesis and transport of serotonin in the VTA of winning mice compared to controls.

A total of 9 out of 44 DEGs encode transcription factors (Table 2), five of which (*Ercc2*, *Lbx1*, *Maf*, *Nkx6-1*, *Tcf7l2*) are associated with the behavior/neurological phenotype and exhibit reduced expression levels in the VTA of winners. An analysis carried out in the Kyoto Encyclopedia of Genes and Genomes (KEGG) database did not reveal metabolic pathways significantly enriched by DEGs.

### 2.2. DEGs in the VTA of Losers vs. Controls

A total of 13,656 expressed genes were identified, among which 188 genes were characterized as differentially expressed in the VTA of defeated mice (losers) compared to controls (Appendix A
Appendix A). Most DEGs, namely 112 genes (59.6%), showed a decreased level of transcription in defeated mice compared to controls. Altogether, 64 genes associated with the behavior/neurological phenotype were found among the DEGs (Table 1). Six of these genes (*Camk2a*, *Egr1*, *En2*, *Fev*, *Gad2*, *Tph2*) are known to be associated with abnormal aggression-related behavior and an increased aggression towards mice. Sixteen genes from the list in Table 1 are associated with abnormal fear/anxiety-related behavior, six of which (*Camk2a*, *Gad2*, *Slc6a2*, *Slc6a3*, *Slc6a4*, *Tph2*) are also known to be involved in abnormal depression-related behavior.

A total of 28 out of 188 DEGs encode transcription factors (Table 2), most of which (16 genes) are expressed at reduced levels in the VTA of defeated mice. Among the DEGs encoding transcription factors, 16 genes are associated with the behavior/neurological phenotype.

The analysis of DEGs detected by comparing the level of gene expression in defeated mice to controls revealed several metabolic pathways (Table 3). The expression characteristics of the DEGs linked with these metabolic pathways are presented in Appendix A
Appendix A. According to the changes in the level of transcription of genes related to the metabolic pathways in defeated animals compared with control animals, the activation of GABAergic synapses should be assumed, which includes the activation of GABA synthesis (*Gad1*, *Gad2*) and GABA receptors (*Gabra1*, *Gabrb2*, *Gabrg2*); in addition, activation of the synthesis (*Th*, *Ddc*) and transport (*Slc6a3*) of dopamine can be assumed. These genes are also implicated in drug (nicotine, amphetamine, cocaine) dependence.

### 2.3. DEGs with a Changed Expression Both in Winners and Losers (Common DEGs)

A comparison of the corresponding DEG lists revealed 23 common genes which significantly changed their expression in the VTA of both winners and losers compared to controls (Table 4).

It can be seen that 22 of these genes either increased or decreased the level of transcription unidirectionally in both experimental groups, and only one gene (*Nrgn*, neurogranin) changed the level of transcription in winners and defeated animals in different directions with respect to the controls. In the VTA of winners, the *Nrgn* gene transcription level was decreased, and in the defeated mice the *Nrgn* gene transcription level was increased compared to control animals. Therefore, it can be concluded that, regardless of whether the animals win or are defeated, the level of transcription of many genes in the VTA of experimental mice changes unidirectionally, which is probably the result of the reaction of the animal organism to conditions of social stress. It can be assumed that genes whose expression level varies in different directions can make the most significant contribution to the formation of alternative (aggressive or depressive) behaviors. According to our results, *Nrgn* is one of such genes.

To identify more genes that changed the transcription levels in the VTA of the winners and losers in opposite directions, a comparative analysis of gene expression in these two experimental groups was performed.

### 2.4. DEGs in the VTA of Winners vs. Losers

When comparing the level of gene expression in the VTA of winners and losers, a total of 207 DEGs (Appendix A
Appendix A) were identified, of which 89 genes (43.0%) had a decreased transcription level in defeated mice.

The significantly enriched metabolic pathways and expression characteristics of the related DEGs are shown in Table 5.

The changes in gene expression levels suggest an activation of the dopaminergic metabolic pathway in the VTA of defeated mice, differences in calcium signaling, and activation of several genes related to the axon guidance metabolic pathway, representing a key stage in the formation of neuronal networks, as well as a decrease in the expression of several extracellular matrix receptor interaction genes.

An analysis of DEG expressions in winners vs. losers, revealed six genes with an expression that level changed in opposite directions relative to controls (Table 6). Four of them encode transcription factors. Several genes (*Ercc2*, *Nrgn*, *Otx2*, and *Tcf7l2*) are annotated as being related to the behavior/neurological phenotype, and *Ercc2* as involved in abnormal emotional/affective behaviors. Among these DEGs (listed in Table 6), there may be the genes whose expression patterns determine behavioral differences between winners and losers.

### 2.5. DEGs that Can Maximize the Differences between the Groups of Winners and Losers

The genes with the most significant contribution to the formation of intergroup differences were sought using a partial-least squares discriminant analysis (PLS-DA). The location of the mice from the experimental groups (winners and losers) in the constructed PLS-DA axes is shown in Figure 1A. Next, a correlation analysis between the obtained coordinates of the mice and the level of gene expression in the winners and losers was performed. A histogram showing the distribution of expressed genes according to the obtained values of correlation coefficients is given in Figure 1B. Genes characterized by maximum correlation coefficients are believed to contribute most significantly to intergroup differences.

For further analysis in the KEGG database, 506 genes characterized by correlation coefficients greater than 0.90 or −0.90 (the list of these genes is presented in Appendix A
Appendix A) were selected. Their analysis revealed the significance of several metabolic pathways (Table 7), among which melanogenesis and calcium signaling pathways are noted as the most significantly enriched. Among the DEGs assigned to the identified metabolic pathways, the most common gene is *Camk2a*, encoding calcium/calmodulin-dependent protein kinase II alpha.

*Camk2a* is one of the 63 DEGs characterized by correlation coefficients greater than 0.90 or −0.90. The detailed description of these DEGs is given in Appendix A
Appendix A. The list of these genes includes 23 genes associated with the behavior/neurological phenotype, six of which (*Ercc2*, *Bcl11b*, *Pitx2*, *Otx2*, *En1*, *Foxp2*) encode transcription factors. Appendix A
Appendix A presents a total of 17 genes encoding transcription factors.

The list of genes presented in Appendix A
Appendix A includes four genes (*Six3*, *Ercc2*, *Otx2*, and *Nrgn*) whose expression levels in the winners and defeated mice changed in opposite directions. DEGs bidirectionally changing the level of transcription in winners and defeated mice and associated with the behavior/neurological phenotype (*Ercc2*, *Otx2*, *Nrgn*) can be considered as the key genes in the formation of alternative behaviors in the conditions of social stress. It should be noted that the most highly expressed of these genes are the *Ercc2* and *Nrgn* genes.

According to the Gene Ontology, the *Ercc2* gene is associated with biological processes such as neurogenesis (gliogenesis), the regulation of transcription from RNA polymerase II promoter (regulation of gene expression), programmed cell death, stem cell differentiation, response to stimulus, response to hypoxia, immune system development, and aging, and the *Nrgn* gene is associated with biological processes such as nervous system development, cell–cell signaling, trans-synaptic signaling, behavior, cognition, the regulation of signaling, learning or memory, response to a stimulus, signal transduction, regulation of synapse structure or activity, modulation of synaptic transmission, regulation of synaptic plasticity, associative learning, and intracellular signal transduction.

A list of 63 DEGs presented in Appendix A
Appendix A was used in STRING database utilities to identify possible associations between the encoded proteins. According to the results presented in Figure 2, the network includes 31 genes (shown in red) encoding proteins associated with nervous system development, 24 of which are associated with the generation of neurons (shown in blue).

Among them, several nodes were identified. One of the main nodes is the protein encoded by the *Camk2a* gene. An analysis revealed the interaction of Nrgn with Camk2a, which in its turn is linked to a group of proteins associated with a number of ionic (calcium and sodium) channels. In addition, Nrgn may be associated with glycine transporters (Slc6a5). In addition to Camk2a, proteins encoded by the *Six3* and *Otx2* genes can also be considered nodal ones. Since the expression of these genes, like the expression of the *Nrgn* gene, is changed in opposite directions in winners and defeated mice compared to controls, they can be considered as potential candidates for further studies.

Thus, we identified four genes (*Nrgn*, *Ercc2*, *Otx2*, and *Six3*) that change the level of transcription in opposite directions in winners and losers, thereby contributing to the formation of intergroup differences, including, possibly, the formation of alternative forms of behavior.

### 2.6. Correlation Between the Expression of Nrgn, Ercc2, Otx2, and Six3 and the Genes Involved in the Synthesis and Transport of Dopamine in the VTA Was Determined by a Cluster Analysis Using Expression Data for the Genes Differentially Expressed in the VTA of Winners and Losers

The results are presented in Appendix A. As can be seen, these four genes and *Th*, *Ddc*, and *Slc6a3* genes encoding tyrosine hydroxylase, DOPA decarboxylase, and dopamine transporters, respectively, are in the same cluster. Therefore, it can be assumed that genes with oppositely changed expressions in winners and losers may be coregulated or functionally associated with the synthesis and transport of dopamine in the VTA.

According to our data, two genes encoding dopamine receptors, *Drd2* and *Drd5*, are expressed in the VTA of mice. Based on the criteria of the Cufflinks/Cuffdiff packages used in our work, the *Drd2* gene was not characterized as a DEG in all the comparisons; however, its transcription level increased 1.55 times in the VTA of defeated mice as compared to controls (*p* < 0.009) and only 1.02 times in the VTA of winners (*p* < 0.91) as compared to controls, which remained practically unchanged. Therefore, it can be assumed that a change in the level of *Drd2* transcription may play a functionally important role in the VTA of defeated mice. The expression level of the *Drd5* gene remained stable in all three experimental groups.

To understand whether changes in the level of expression of the genes characterized above as key ones can be related to alternative dopaminergic signaling cascades in winners and losers, the correlation between their level of expression and expression levels of the *Drd2*, *Th*, *Ddc*, and *Slc6a3* genes was calculated in both experimental groups (Table 8). The results showed a strong correlation between the expression of *Nrgn* and the above four genes in defeated animals. In contrast to the group of winners, a statistically significant correlation between the expression of *Nrgn* and *Ercc2* and dopaminergic genes was obtained in the VTA of defeated animals. Three nodal genes (*Camk2a*, *Otx2*, and *Six3*) were characterized by high correlation coefficients with the dopaminergic genes in the VTA of mice from both experimental groups.

## 3. Discussion

First of all, when discussing the results of this study, it is important to comment on the alterations in the expression levels of the genes involved in the synthesis or transport of neurotransmitters. The analysis showed that the *Tph2* and *Slc6a4* genes responsible for the synthesis and transport of serotonin, respectively, in the VTA are expressed at higher levels in both winners and losers compared to the controls. The role of serotonin in both aggression and anxiety/depression has been extensively studied and reviewed by many authors [18,19,20]. Although serotonin is believed to be a potent inhibitor of aggressive behavior, there are numerous data in favor and against this hypothesis (reviewed in [21]). The complexity of the mechanisms regulating serotonin-mediated behavior is noted in many studies [19,22]. It is reported that the pharmacological effects of drugs depend on age, environment or experimental conditions [18,21]. Therefore, when discussing the results of this study, one should bear in mind that they are mostly relevant to the development of mixed anxiety–depressive disorder and pathological aggressive behavior in mice within the framework of the used experimental model.

In the current study, in the experimental groups of mice with alternative types of behavior, unidirectional changes in the expression of serotonergic genes indicate that the synthesis and transport of serotonin are activated as a reaction of the serotonergic system to experimentally induced social stress experienced by animals and probably are not associated with the formation of alternative forms of behavior. This view is in good agreement with the opinion of other authors who believe that changes in serotonin levels associated with abnormal behavior are conditioned by different factors [19]. It should also be noted here that the defeated animals are likely to experience more severe stress as indicated by a significantly higher increase in the transcription of serotonergic genes in the VTA of defeated mice than in the winners compared to controls (Table 4).

As already noted in the introduction, most VTA neurons are dopaminergic. Statistically significant increases in the transcription levels of several genes (*Ddc*, *Slc6a3*, *Th*) controlling dopamine synthesis (*Th*, *Ddc*) and transport (*Slc6a3*) were detected in the VTA of defeated mice as compared to controls. In winning mice, the respective changes were less significant. Compared to controls, in the VTA of winning animals the increase in the level of transcription of the *Ddc* gene was characterized by a *p* value = 0.0028, whereas for *Slc6a3* and *Th* it was statistically insignificant according to the annotations of the Cuffdiff program. At the same time, we previously studied the expression of these genes in larger groups of animals by real-time PCR and showed that in the VTA of the winning mice the level of transcription of the *Slc6a3* and *Th* genes was significantly increased [23]. Therefore, it can be assumed that in the VTA of both winning (aggressive) and defeated mice, dopamine synthesis increases, however, as in the case of serotonin, the effect of stress on dopamine synthesis and transport in defeated mice is stronger than that in the winners.

GABAergic neurons also play an important role in VTA functioning [14]. Several genes (*Gabra1*, *Gabrb2*, *Gabrg2*, *Gad1*, *Gad2*) related to the functioning of GABAergic synapses demonstrated a significantly increased transcription in defeated mice compared to controls (Table 4 and Appendix A
Appendix A). According to our RNA-Seq data, the level of transcription of all these five genes also increased in the group of aggressive animals, but the differences compared to the control did not reach statistically significant values. Accordingly, we can assume both an increase in the synthesis of GABA and an increase in the expression of its receptors (*Gabra1*, *Gabrb2*, *Gabrg2*) in both experimental groups of mice, but in aggressive animals, changes in the function of GABAergic synapses were less pronounced than in the defeated ones. The possible functional consequences of significant changes in these genes in defeated animals were discussed in [9]. Here we want to emphasize that the unidirectional activation of a number of genes that control the functioning of GABAergic neurons occurs in the VTA of mice of both experimental groups and that this activation is more pronounced in defeated animals. The fact that defeated mice are more stressed than winners also follows from almost four times the number of DEGs that were found in losers vs. controls than in winners vs. controls comparisons. These results are in very good agreement with our observation that both winners and defeated mice are characterized by increased anxiety [6,24,25,26].

It should be noted that in the current study the social stress experienced by animals caused a change in the expression level of multiple genes that encode transcription factors in both experimental groups of mice, suggesting a wide range of functional changes in the VTA and, with a high probability, in the areas of the brain onto which the VTA neurons project. As can be seen from Table 4, which represents common DEGs, three DEGs encoding transcription factors (*Lbx1*, *Maf*, *Nkx6-1*) reduced the level of transcription in both winners and defeated mice, therefore suggesting that these genes may play a key role in regulating the overall response to stress.

Almost all genes listed in Table 4 have their transcription levels changed in the same direction in both winners and defeated mice, and only one (*Nrgn*) has it changed in the opposite direction—its transcription level decreased in winners and increased in defeated mice. The *Nrgn* gene encodes neurogranin (Ng), which is enriched at dendritic spines and can enhance synaptic strength by targeting calmodulin (CaM) [27]. The bidirectional changes in the level of *Nrgn* transcription found in our experiment are in good agreement with the fact that neurogranin is associated with both long-term potentiation and long-term depression [28]. Thus, it can be assumed that this particular gene can be involved in the formation of alternative behaviors in the experimental groups of mice.

In physiological conditions, neurogranin forms a complex with CaM, and its CaM-binding affinity can be modulated by phosphorylation, oxidation and glutathiolation under the activation of protein kinase C or oxidant stress [29]. It was shown that mice with an *Nrgn* gene deletion displayed an apparently normal phenotype but were characterized by impairments in spatial and emotional learning and by changes in hippocampal short- and long-term plasticity (paired-pulse depression, synaptic fatigue, and long-term potentiation induction). These deficits were accompanied by a decreased basal level of the activated Ca(2+)/CaM-dependent kinase II (CaMKII) [30,31]. The behavioral testing revealed that neurogranin knockout mice were both hyperactive and socially withdrawn [32]. Neurogranin overexpression in neurons enhances synaptic strength as well as long-term potentiation (LTP) and learning by promoting calcium-mediated signaling [33]. Based on the above information, we can assume that the increase in *Nrgn* transcription in the VTA of defeated mice and its decrease in the VTA of winners observed in the current study are adaptive reactions of the system working to achieve a homeostatic balance.

It was shown that the restorative effects of neurogranin on synaptic depression and LTP deficits are dependent on the interaction of neurogranin and CaM and CaM-dependent activation of CaMKII [34]. The results of our experiment showed a coordinated increase in the expression of both *Nrgn* and *Camk2a* (encoding calcium/calmodulin-dependent protein kinase II alpha) genes in the VTA of defeated mice, while the level of *Camk2a* transcription was unchanged in the VTA of the winners exhibiting a decreased level of *Nrgn* transcription.

CaMKII is known as a mediator of Ca^2+^-linked signaling. It phosphorylates a wide range of substrates playing a multifunctional role in the coordination and regulation of Ca^2+^-mediated alterations in different intracellular events related to the neuronal functions: the synthesis and release of neurotransmitters, modulation of ion channel activity, cellular transport, cell morphology and neurite extension, synaptic plasticity, gene expression, etc. [35,36,37]. *Camk2a* encodes a calcium/calmodulin-dependent protein kinase subunit that plays a key role in synaptic plasticity, AMPA receptor transmission, LTP, and long-term memory formation, and its dysfunction underlies neuropsychiatric disorders such as drug addiction, schizophrenia, depression, epilepsy, and multiple neurodevelopmental disorders, perhaps through maladaptations in glutamate signaling and neuroplasticity [38,39]. The data obtained in the current study are in a good agreement with the findings that calcium signaling is an important link in the regulation of processes associated with neurogenesis in various neurological disorders [40].

In our work, when comparing the transcriptomes of the winning and defeated mice, the *Camk2a* gene was assigned to the DEGs with maximum contributions to the intergroup differences (Appendix A
Appendix A). Figure 2 clearly demonstrates that the protein encoded by the *Camk2a* gene can be viewed as a nodal gene in the calcium modulating pathway, as well as in the regulation of sodium and potassium ion channels and a number of other processes responsible for functional differences in the VTA of winners and defeated animals.

The list of DEGs characterized as making a maximum contribution to intergroup differences contains three more genes (*Ercc2*, *Otx2*, *Six3*) that change the transcription level in winners and defeated mice in opposite directions. These three genes encode transcription factors.

The *Ercc2* gene is associated with the behavior/neurological phenotype. In the VTA, a decrease in the level of transcription of this gene was detected in the winners, and an increase in its transcription was detected in the defeated mice (Table 6). Given that *Ercc2* is involved in repairing the damage caused by the redox process [41], it can be assumed that oxidative stress processes in the VTA play a pivotal role in the formation of phenotypic features of defeated mice.

According to the results presented in Figure 2, the proteins encoded by the transcription factor genes *Otx2* and *Six3*, the expressions of which are changed in opposite directions in winners and defeated mice compared to controls, can also be considered as key (nodal) genes. The investigations of the mechanisms involved in neural functions have shown that the expression of several regulatory genes, including *Six3*, cannot be initiated in the neural plate of Otx2−/− embryos [42,43], suggesting that the expressions of the *Otx2* and *Six3* genes are closely related.

Recently the *Otx2* gene was characterized as an upstream mediator of increased susceptibility to social defeat stress in mice. Its transient knockdown in the VTA increased stress susceptibility, and its overexpression was associated with reverse effects [44]. The role for the *OTX2* gene was confirmed in the study of stress-related depressive disorder pathophysiology in children [45]. Otx2 is involved in the regulation of the number of VTA neurons with efficient dopamine uptake [46] and is considered one of the genes with a key role in midbrain dopaminergic neuron development and the regulation of their survival and physiology [47].

To date, the *Six3* gene has not been associated with the behavior/neurological phenotype, but its role in the control of the status of the neural progenitor cells has been reported. It was shown that the upregulation of *Six3* expression kept the progenitor cells of the embryonic telencephalon in an undifferentiated state. The authors suggested that *Six3* is involved in the control of the subtle equilibrium between the proliferation and differentiation of neural progenitor cells during mammalian neurogenesis [48]. A conditional deletion of the *Six3* gene prevented the formation of most dopamine receptor DRD2-expressing medium spiny neurons being the principal projection neurons in the striatum [49]. To date, the role of the *Six3* gene in the functioning of the VTA neurons has not been reported. However, based on the above-cited data on its role in the formation of the DRD2-expressing neurons of the striatum, it can be assumed that the *Six3* gene can also participate in the regulation of D2R-expressing neurons functioning in the VTA. Our results from the cluster and correlation analyses confirm the plausibility of this hypothesis. Accordingly, a further study of the effects of *Six3* gene expression in the VTA of the winners and defeated animals on dopaminergic signaling may be promising for understanding the formation of their behavioral characteristics.

The VTA is one of the major sites of dopamine synthesis. The VTA dopaminergic neurons are implicated in the regulation of emotional and motivational behaviors and are involved in the development of psychopathologies including depression [50], aggression [51], and addictions [52,53]. So, it was important to check if the prioritized genes could be implicated in VTA dopaminergic signaling. The results of the cluster analysis on expression data of the genes differentially expressed in the VTA of winners and defeated mice showed that the *Nrgn*, *Ercc2*, *Otx2*, *Six3*, and *Camk2a* genes were assigned to the same cluster as the dopaminergic (*Th*, *Ddc*, *and Slc6a3*) genes. According to the correlation analysis, the expression of *Otx2*, *Six3*, and *Camk2a* genes is coregulated with the dopaminergic genes in both experimental groups. The expression of two other genes (*Nrgn* and *Ercc2*) correlated with the expression of dopaminergic genes only in the VTA of losers indicating a possible coregulation or functional relationship of these genes in the VTA of defeated mice. It should be emphasized that the most significant correlations were found between the expression of dopaminergic genes and the expression of the *Nrgn* gene (see Table 8).

The development of technologies generating large amounts of gene expression data called for the use of mathematical methods capable of grouping the genes based on the similarity of their expression patterns. Such methods include a correlation analysis [54,55], which is a potent tool to identify coregulated genes or groups of genes associated with particular experimental or environmental conditions [56].

To date, the role of the *Nrgn* gene in dopaminergic signaling remains unclear, however, there is indirect evidence for its involvement [57]. Additionally, in favor of this hypothesis are the findings that neurogranin is enriched at dendritic spines [27] whereas D2R is localized to the VTA dendrites [58], which suggests their spatial colocalization.

According to our results, *Nrgn* is involved in the network of 31 coregulated genes, encoding proteins associated with the nervous system development, including 24 genes encoding proteins associated with the generation of neurons. Several of them (Otx2, En1) are known as regulators of terminal differentiation, survival and the maintenance of midbrain dopaminergic neurons [59]. We believe that other coregulated genes included in the network may also be essential for studying their role in VTA dopaminergic neuron maturation under the influence of social stress.

Of course, many genes with VTA expression levels significantly changed in one of experimental groups and not changed in the other group could be directly or indirectly related to the formation of behavioral characteristics in winners and defeated mice. However, we believe that the genes that change the level of transcription in opposite directions in winners and defeated mice deserve special research attention. As noted earlier, the *Nrgn*, *Ercc2*, *Otx2*, and *Six3* genes that change their transcription levels in opposite directions in winners and defeated mice and make the maximum contribution to intergroup differences can be considered as key genes in the formation of alternative phenotypes associated with behavioral patterns in winning and defeated animals.

## 4. Materials and Methods

### 4.1. Animals

The work was carried out using 10–12-week old C57BL/6J male mice obtained from the Animal Breeding Facility, Branch of Institute of Bioorganic Chemistry of the Russian Academy of Sciences (Pushchino, Moscow region, Russia). Mice were kept under standard conditions at 22+/−2 °C with a 12/12 h light–dark cycle (light period started from 8:00 AM) and dry laboratory food and water given ad libitum. All procedures were in compliance with the European Communities Council Directive 210/63/EU on 22 September 2010. The study was approved by Scientific Council N 9 of the Institute of Cytology and Genetics SB RAS of 24 March 2010, N 613 (Novosibirsk, Russia).

### 4.2. Generation of Alternative Forms of Social Behavior in Male Mice under Agonistic Interactions

Alternative forms of social behavior were generated as a result of the daily agonistic interactions (intermale confrontations) of male mice as described previously [1,3]. Pairs of weight-matched mice were each placed in a cage (14 × 28 × 10 cm) bisected by a perforated transparent partition allowing the animals to see, hear and smell each other, but preventing physical contact. The animals were left undisturbed for two or three days to adapt to unfamiliar housing conditions before they were exposed to encounters. Every day at 14:00–17:00 p.m. (Russian local time), the cage lid was replaced by a transparent one, and after 5 min (the period necessary to activate agonistic interactions), the partition was removed for 10 min for intermale confrontation. The mouse that attacked, bit, and chased the opponent was considered the winner. The superiority of the winner was established by the results of two or three encounters with the same opponent. A mouse that showed only defensive behavior (sideways postures, upright postures, withdrawal, lying on the back or freezing) was defined as the loser. If the aggressive attacks were very active and long, then the interactions between the males were stopped after 3 min (or even earlier) by restoring the partition, to prevent damage to the defeated male. That means that the painful effects of agonistic interactions are absent in this model for defeated mice.

Each defeated mouse (loser) was exposed to the same winner for three days, and then, to continue the agonistic interactions, the defeated mouse was placed in an unfamiliar cage with an unfamiliar winner behind the partition. Each winner remained in its original cage. The intermale confrontations procedure was performed once a day for 21 days and yielded an equal number of winners and losers.

Three groups of animals were used: (1) controls—mice without a consecutive experience of agonistic interactions; (2) winners—group of aggressive mice chronically winning during 21 days in daily agonistic interactions (intermale confrontations); (3) losers (defeated mice)—mice with chronic experience of defeats during 21 days in daily agonistic interactions. Animals with alternative types of social experience demonstrated the development of various pathological behaviors. Mice with a long experience of aggression and victories (winners) showed increased aggressiveness, hyperactivity, stereotyped behaviors, anxiety, impaired social recognition, irritability, autistic spectrum symptoms, a condition similar to drug addiction, etc. (reviewed in [60]). The losers developed mixed anxiety/depression-like behaviors accompanied by full immobility, strong anxiety, avoidance of any social interactions, helplessness, indifference, etc. [6,61].

A group of 21 pairs of male C57BL/6J mice was used to generate alternative forms of social behavior. Winners and losers with the most expressed behavioral phenotypes were selected for the transcriptome analysis. The following criteria were used. For the losers, during the activation period (5 min before a fight) the chronically defeated mice demonstrated all symptoms of depressive behavior: they did not approach the partition, sit in the corner of the cage opposite to the partition, or in a corner or into litter with their nose; they were characterized by immobility, freezing under winners’ attacking, and a demonstrated indifference in all experimental situations (without behavioral reactions); there were no inversions of behavior to the opposite one after a change of aggressors, and they showed avoidance and passive defense when attacked by the aggressor. Aggressive winners, during the activation period demonstrated a strong aggressive motivation, and every day immediately attacked the losers after partition removal, stopping only for rest, displaying a manic motivation to bite losers in spite of full submission.

Control animals and all experimental mice were simultaneously decapitated. Experimental mice were decapitated 24 h after the last agonistic interaction. The brain regions were dissected by the same experimenter according to the map presented in the Allen Mouse Brain Atlas [62]. All biological samples were placed in RNAlater solution (Thermo Fisher Scientific, Waltham, MA, USA) and stored at −70 °C until sequencing.

### 4.3. RNA-Seq Analysis

The frozen VTA samples from male mice: winners (*n* = 3), losers (*n* = 3) and controls (*n* = 3), were sent to JSC Genoanalytica (Moscow, Russia) which specializes in RNA-Seq analysis. For transcriptome profiling, mRNA was extracted using the Dynabeads mRNA Purification Kit (Ambion, Thermo Fisher Scientific, Waltham, MA, USA). cDNA libraries were constructed using the NEBNext mRNA Library Prep Reagent Set for Illumina (NEB, Ipswich, MA USA). All experimental procedures were performed in accordance with the manufacturer’s protocols. A single-end sequencing of cDNA libraries was performed on the Illumina Hiseq 1500 platform (Illumina Sequencing, San Diego, CA, USA) with a read length of 50 bases. All samples were analyzed as biological replicates. The quality metrics of the mapped data (Appendix A
Appendix A) were collected using Spliced Transcripts Alignment to a Reference (STAR) software (Cold Spring Harbor, NY, USA) [63]. The sequencing data were mapped to the mouse reference genome sequence (GRCm38.p3) available in GenBank using the TopHat2 aligner (Center for Bioinformatics and Computational Biology, University of Maryland, College Park, MD, USA) [64]. The RNA-Seq data sets are available at European Nucleotide Archive (Accession PRJEB39532)

Cufflinks/Cuffdiff programs were run to estimate the gene expression levels in FPKM (fragments per kilobase of transcript per million mapped reads) and to identify the differentially expressed genes (DEGs) in the experimental and control groups. Genes were considered differentially expressed at a false discovery rate (q value) <5% [65]. Only annotated gene sequences were used in the analysis.

DEGs obtained by comparing winners with losers were considered as bidirectionally changing the level of expression if their transcription level changed in different directions by at least 1.5 times relative to the control in the control/winner and control/loser comparisons.

### 4.4. Functional Annotation

The functional annotation of DEGs was performed using DAVID (The Database for Annotation, Visualization and Integrated Discovery) gene annotation tool [66]. The Mus musculus genome was utilized as the background list for the over-representation analysis. The Gene Ontology (GO) option in DAVID and Kyoto Encyclopedia of Genes and Genomes (KEGG) Pathway Database [67] were used to identify the significantly (*p* < 0.05) enriched biological processes and metabolic pathways. To construct the functional protein association network the STRING database was used [68].

To define the association of DEGs with behavior/neurological phenotype the Neurological Disease Portal (Phenotypes, Mouse) in Rat Genome Database (RGD [69] was employed.

An atlas of combinatorial transcriptional regulation in mice and men [70] was used to reveal the DEGs encoding the transcription factor genes.

### 4.5. Statistical Methods

The acquired RNA-Seq data (in FPKM values) were log transformed, centered, and normalized. Then, the data sets were scaled by the principal coordinates (PCOs) method based on Euclidean metric distances and the partial-least squares discriminant analysis (PLS-DA) was conducted using the pattern of covariation for linear combinations between two blocks of variables [71]. The constructed PLS-DA Axis maximized the distance between the analyzed groups of animals. After that, the Pearson correlation between gene expression levels and the coordinates of animals along the first functionally meaningful synthetic axis (PLS-DA Axis 1) was employed to define a set of expressed genes that are characterized by the highest correlation coefficients. Such genes are thought to contribute the most to intergroup differences. Normalized RNA-Seq data were also used for correlation analysis. The software packages Statistica 6.0 (StatSoft, Tulsa, OK, USA) and JACOBI4 [72] were used for the multivariate analysis and representation of the data.

Agglomerative Hierarchical Clustering (AHC) analysis was carried out by XLStat software -(Version 2016.02, Addinsoft Inc., Paris France). We used the unweighted pair-group average agglomerative method, noncentered mode, with the Pearson pairwise correlation matrix as an input.

## 5. Conclusions

The analysis of gene expression profiles in the VTA of mice with alternative social behaviors showed that during daily social confrontations, the transcription of many genes in both chronically defeated and aggressive mice changes unidirectionally, which indicates that mice in both experimental groups are under stress. However, the results unambiguously indicated that the consequences of long-term social stress of agonistic interactions are much more pronounced in defeated mice, which develop a mixed anxiety/depression-like state under conditions of chronic unavoidable psychoemotional social stress.

The comprehensive bioinformatics analysis carried out in this work made it possible to identify the potentially important candidate genes (*Nrgn*, *Ercc2*, *Otx2*, and *Six3*) that may be involved in the formation of the behavioral differences in losers and winners. It was first shown that *Nrgn* (neurogranin) expression was highly correlated with the expression of the genes involved in dopamine synthesis and transport (*Th*, *Ddc*, *Slc6a3*, and *Drd2*) in the VTA of defeated mice but not in winners. This study may be useful for understanding specific and nonspecific molecular events occurring in VTA neurons during the formation of human disorders associated with aggressive or depressive-like states.

The limitations of this study include the following:It is necessary to remember the complexity of the mechanisms regulating behavior. Accordingly, the results obtained in this study may be valid for mixed anxiety/depressive disorder and pathological aggressive behavior in mice within the framework of the used experimental model. However, we tried to describe the methodology of the experiment in detail in order to increase the replicability of the study.The sequencing and bioinformatic analysis carried out in this work is only the beginning of the process of studying the role of the identified candidate genes (*Nrgn* and other highlighted genes) in the formation of behavioral features under the influence of social confrontations. To establish the relationship between the transcription of the candidate genes and the external (behavioral) phenotype, additional studies at the proteome and metabolome levels are required.

## Figures and Tables

**Figure 1 ijms-21-06599-f001:**
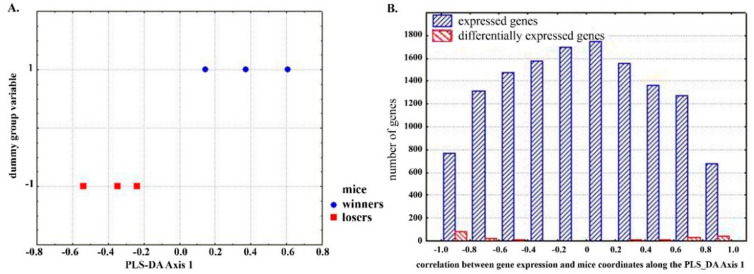
(**A**). Axis maximizing the distances between winning and defeated mice; (**B**). The distribution of expressed genes along the axis representing the correlation between gene expression and mice coordinates along the partial-least squares discriminant analysis (PLS-DA) Axis 1.

**Figure 2 ijms-21-06599-f002:**
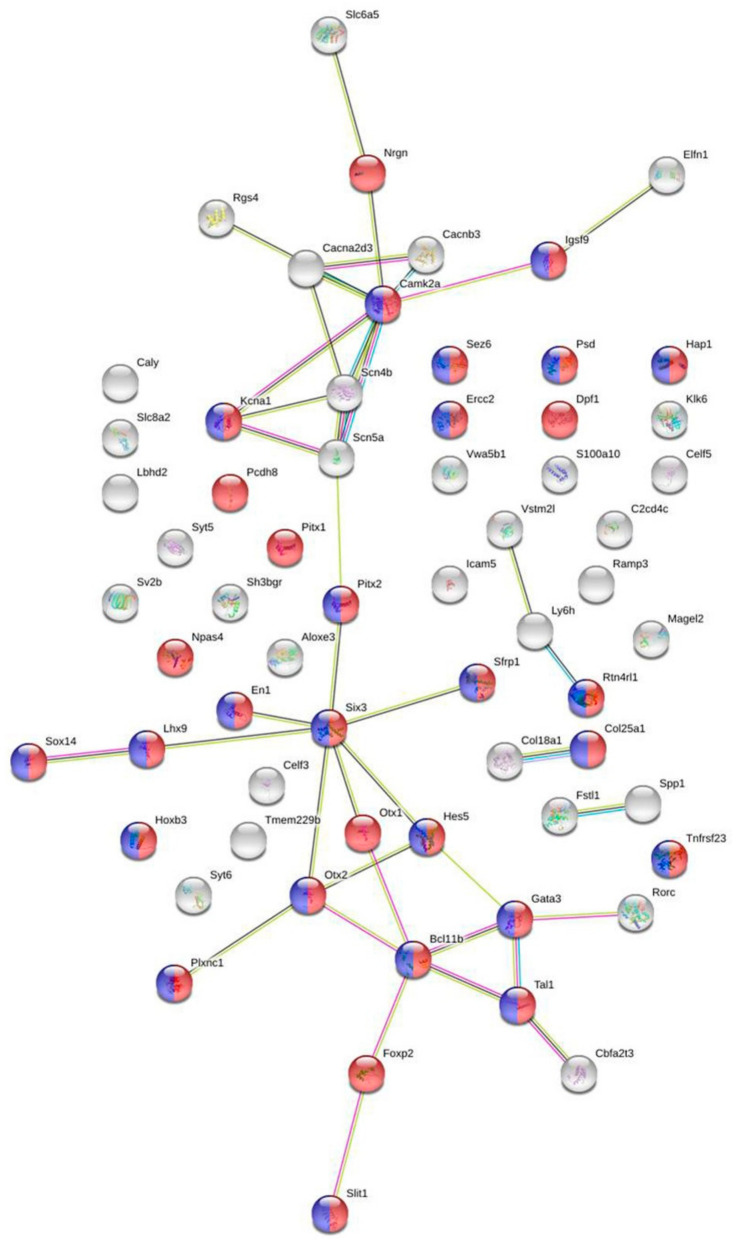
The relationships between DEGs that make the most significant contribution to the intergroup (winners vs. losers) differences according to the PLS-DA analysis. STRING database utilities were used to identify possible associations between encoded proteins. The genes encoding proteins associated with the nervous system development are shown in red; the genes associated with generation of neurons are shown in blue.

**Table 1 ijms-21-06599-t001:** Differentially Expressed Genes (DEGs) associated with the behavior/neurological phenotype.

Gene Symbol	Locus	Expression in VTA (FPKM)		Log2 (Fold Change) Winners/Controls	q_Value	Definition
		Control Mice	Winners			
Winners vs. Control Mice						
*Dnaja1*	chr4:40720153-40757885	41.90	104.77	1.32	2.05 × 10^−2^	DnaJ heat shock protein family (Hsp40) member A1
*Ercc2* ^#^	chr7:19382024-19404104	53.42	8.72	−2.62	2.05 × 10^−2^	excision repair cross-complementing rodent repair deficiency, complementation group 2
*Lbx1*	chr19:45232619-45235377	7.18	3.19	−1.17	2.05 × 10^−2^	ladybird homeobox homolog 1 (Drosophila)
*Maf*	chr8:115682935-115707940	9.66	5.98	−0.69	3.58 × 10^−2^	avian musculoaponeurotic fibrosarcoma oncogene homolog
*Myh6*	chr14:54936101-54966607	1.47	59.90	5.35	2.05 × 10^−2^	myosin, heavy polypeptide 6, cardiac muscle, alpha
*Ngfr*	chr11:95568820-95587698	3.55	2.01	−0.82	2.05 × 10^−2^	nerve growth factor receptor (TNFR superfamily, member 16)
*Nkx6-1*	chr5:101658055-101665361	5.01	2.79	−0.85	3.58 × 10^−2^	NK6 homeobox 1
*Nptx2*	chr5:144545886-144557478	3.28	1.59	−1.04	2.05 × 10^−2^	neuronal pentraxin 2
*Nrgn*	chr9:37544492-37552745	22.35	14.15	−0.66	3.58 × 10^−2^	neurogranin
*Slc17a7* ^#,§,©^	chr7:45163920-45176328	10.74	20.59	0.94	2.05 × 10^−2^	solute carrier family 17 (sodium-dependent inorganic phosphate cotransporter), member 7
*Slc6a4* ^#,§,©^	chr11:76998596-77032343	5.03	9.99	0.99	2.05 × 10^−2^	solute carrier family 6 (neurotransmitter transporter, serotonin), member 4
*Sox18*	chr2:181669836-181671640	9.60	5.15	−0.90	2.05 × 10^−2^	SRY (sex determining region Y)-box 18
*Tcf7l2*	chr19:55741714-55933693	6.56	3.44	−0.93	2.05 × 10^−2^	transcription factor 7 like 2, T cell specific, HMG box
*Tph2* ^#,∆,¶,§,©^	chr10:115078553-115185022	7.52	13.79	0.87	4.87 × 10^−2^	tryptophan hydroxylase 2
**Losers vs. Control Mice**						
**Gene Symbol**	**Locus**	**Control Mice**	**Losers**	**Log2 (Fold Change) Losers/Controls**	**q_Value**	**Definition**
*Akap12*	chr10:4266328-4359605	30.74	19.84	−0.63	0.0159	A kinase (PRKA) anchor protein (gravin) 12
*Anxa4*	chr6:86736652-86793645	5.40	2.37	−1.19	0.0063	annexin A4
*Bag3*	chr7:128523582-128546979	21.99	9.86	−1.16	0.0063	BCL2-associated athanogene 3
*Cacna1e* ^#,§^	chr1:154390344-154884412	7.26	11.61	0.68	0.0275	calcium channel, voltage-dependent, R type, alpha 1E subunit
*Calca*	chr7:114631477-114636357	23.97	5.14	−2.22	0.0063	calcitonin/calcitonin-related polypeptide, alpha
*Camk2a* ^#,∆,¶,§,©^	chr18:60925325-60988993	21.04	32.70	0.64	0.0275	calcium/calmodulin-dependent protein kinase II alpha
*Cdkn1a*	chr17:29090985-29100722	5.82	3.43	−0.76	0.0275	cyclin-dependent kinase inhibitor 1A (P21)
*Cdkn1c*	chr7:143458338-143461074	5.59	3.08	−0.86	0.0063	cyclin-dependent kinase inhibitor 1C (P57)
*Col25a1*	chr3:130180797-130599886	3.25	5.24	0.69	0.0205	collagen, type XXV, alpha 1
*Colq*	chr14:31523083-31577383	2.14	0.78	−1.45	0.0063	collagen-like tail subunit (single strand of homotrimer) of asymmetric acetylcholinesterase
*Dao* ^#,§^	chr5:114003734-114025676	5.17	2.55	−1.02	0.0063	D-amino acid oxidase
*Ddc* ^#,§^	chr11:11814099-11898144	14.29	35.12	1.30	0.0063	dopa decarboxylase
*Dnaja1*	chr4:40720153-40757885	41.15	155.22	1.92	0.0063	DnaJ heat shock protein family (Hsp40) member A1
*E2f1*	chr2:154559399-154569892	4.75	2.38	−0.99	0.0063	E2F transcription factor 1
*Egr1* ^#,∆^	chr18:34861206-34864956	9.62	15.75	0.71	0.0115	early growth response 1
*En1* ^#,§^	chr1:120602486-120607991	5.32	10.27	0.95	0.0063	engrailed 1
*En2* ^#,∆^	chr5:28165695-28172166	3.09	5.95	0.94	0.0063	engrailed 2
*Fev* ^#,∆,¶,§^	chr1:74881508-74885408	1.25	3.95	1.66	0.0063	FEV (ETS oncogene family)
*Foxp2* ^#^	chr6:14901169-15442450	1.59	3.15	0.98	0.0063	forkhead box P2
*Gabra1* ^#,§^	chr11:42130920-42183066	45.26	77.94	0.78	0.0063	gamma-aminobutyric acid (GABA) A receptor, subunit alpha 1
*Gabrb2*	chr11:42419746-42632694	19.55	29.08	0.57	0.0446	gamma-aminobutyric acid (GABA) A receptor, subunit beta 2
*Gabrg2* ^#,§^	chr11:41909958-42000874	25.14	38.21	0.60	0.0159	gamma-aminobutyric acid (GABA) A receptor, subunit gamma 2
*Gad1*	chr2:70489939-70602029	48.93	79.73	0.70	0.0063	glutamate decarboxylase 1
*Gad2* ^#,∆,§,©^	chr2:22622057-22694078	34.04	54.29	0.67	0.0243	glutamic acid decarboxylase 2
*Gjc2*	chr11:59175476-59183213	41.32	24.81	−0.74	0.0115	gap junction protein, gamma 2
*Inhbb*	chr1:119415464-119422248	3.44	2.15	−0.68	0.0374	inhibin beta-B
*Irs4*	chrX:141710996-141725254	1.76	3.33	0.92	0.0063	insulin receptor substrate 4
*Isl1*	chr13:116298269-116309693	1.61	0.49	−1.72	0.0063	ISL1 transcription factor, LIM/homeodomain
*Lbx1*	chr19:45232619-45235377	7.04	0.71	−3.32	0.0063	ladybird homeobox homolog 1 (Drosophila)
*Maf*	chr8:115682935-115707940	9.48	5.55	−0.77	0.0063	avian musculoaponeurotic fibrosarcoma oncogene homolog
*Mafb*	chr2:160363676-160367065	4.47	2.51	−0.84	0.0063	v-maf musculoaponeurotic fibrosarcoma oncogene family, protein B (avian)
*Mecom*	chr3:29951264-30013204	1.23	0.60	−1.04	0.0063	MDS1 and EVI1 complex locus
*Mmp9*	chr2:164948218-164955849	2.26	0.89	−1.34	0.0063	matrix metallopeptidase 9
*Myh6*	chr14:54936101-54966607	1.44	20.92	3.86	0.0243	myosin, heavy polypeptide 6, cardiac muscle, alpha
*Nefh*	chr11:4938562-5042794	287.73	119.70	−1.27	0.0063	neurofilament, heavy polypeptide
*Nefl*	chr14:68082365-68092351	914.72	422.75	−1.11	0.0063	neurofilament, light polypeptide
*Nefm*	chr14:68119465-68131373	690.57	269.10	−1.36	0.0063	neurofilament, medium polypeptide
*Neurod2*	chr11:98325416-98329645	2.92	1.59	−0.88	0.0243	neurogenic differentiation 2
*Ngb*	chr12:87097530-87102539	9.25	20.21	1.13	0.0063	neuroglobin
*Ngfr*	chr11:95568820-95587698	3.48	1.91	−0.86	0.0159	nerve growth factor receptor (TNFR superfamily, member 16)
*Nkx6-1*	chr5:101658055-101665361	4.91	1.66	−1.56	0.0063	NK6 homeobox 1
*Nrgn*	chr9:37544492-37552745	21.95	42.78	0.96	0.0063	neurogranin
*Ntng2*	chr2:29193828-29253707	25.61	14.93	−0.78	0.0275	netrin G2
*Otx2*	chr14:48657676-48667644	0.73	2.03	1.49	0.0115	orthodenticle homeobox 2
*Pax7*	chr4:139737093-139832968	0.45	1.64	1.85	0.0063	paired box 7
*Phox2b*	chr5:67094327-67099126	7.03	1.56	−2.17	0.0063	paired-like homeobox 2b
*Prss12* ^#,§^	chr3:123446912-123506602	4.23	1.89	−1.16	0.0063	protease, serine 12 neurotrypsin (motopsin)
*Pvalb* ^#,§^	chr15:78191044-78206351	119.09	70.07	−0.77	0.0063	parvalbumin
*Scn3a*	chr2:65456990-65567519	3.86	6.64	0.78	0.0346	sodium channel, voltage-gated, type III, alpha
*Slc17a8*	chr10:89574019-89621249	0.90	2.12	1.24	0.0063	solute carrier family 17 (sodium-dependent inorganic phosphate cotransporter), member 8
*Slc18a3*	chr14:32407354-32466004	33.83	13.30	−1.35	0.0063	solute carrier family 18 (vesicular monoamine), member 3
*Slc6a2* ^#,§,©^	chr8:92960641-93001667	7.52	4.87	−0.63	0.0346	solute carrier family 6 (neurotransmitter transporter, noradrenalin), member 2
*Slc6a3* ^#,§,©^	chr13:73536746-73578672	1.03	9.51	3.21	0.0063	solute carrier family 6 (neurotransmitter transporter, dopamine), member 3
*Slc6a4* ^#,§,©^	chr11:76998596-77032343	4.94	22.71	2.20	0.0063	solute carrier family 6 (neurotransmitter transporter, serotonin), member 4
*Slc6a5*	chr7:49909938-49963861	45.06	13.79	−1.71	0.0063	solute carrier family 6 (neurotransmitter transporter, glycine), member 5
*Spp1*	chr5:104435110-104441053	54.80	23.57	−1.22	0.0063	secreted phosphoprotein 1
*Syt1*	chr10:108497647-109010983	51.32	86.93	0.76	0.0063	synaptotagmin I
*Tgfb3*	chr12:86056580-86079159	6.99	4.46	−0.65	0.0374	transforming growth factor, beta 3
*Th* ^#,§^	chr7:142892670-142901960	10.93	21.96	1.01	0.0063	tyrosine hydroxylase
*Tph2* ^#,∆,¶,§,©^	chr10:115078553-115185022	7.40	26.97	1.87	0.0063	tryptophan hydroxylase 2
*Vdr* ^#^	chr15:97854426-97908296	0.76	0.29	−1.41	0.0115	vitamin D receptor
*Vgf*	chr5:137025191-137033857	119.24	78.61	−0.60	0.0473	VGF nerve growth factor inducible
*Vipr2*	chr12:116077725-116146261	1.83	0.39	−2.24	0.0063	vasoactive intestinal peptide receptor 2
*Whrn*	chr4:63414854-63496130	21.72	13.85	−0.65	0.0275	whirlin

Genes associated with: ^#^—abnormal emotion/affect behavior; ^∆^—abnormal aggression-related behavior; ^¶^—increased aggression towards mice; **^§^**-abnormal fear/anxiety-related behavior; ©—abnormal depression-related behavior.

**Table 2 ijms-21-06599-t002:** DEGs encoding transcription factors.

Gene Symbol	Locus	Expression in VTA (FPKM)		log2 (Fold Change) Winners/Controls	q_Value	Definition
		Control Mice	Winners			
Winners vs. Control Mice						
*Ddx5*	chr11:106780156-106818861	121.77	381.38	1.65	2.05 × 10^−2^	DEAD (Asp-Glu-Ala-Asp) box polypeptide 5
*Ercc2 **	chr7:19382024-19404104	53.42	8.72	−2.62	2.05 × 10^−2^	excision repair cross-complementing rodent repair deficiency, complementation group 2
*Fubp3*	chr2:31572650-31617590	11.21	60.26	2.43	2.05 × 10^−2^	far upstream element (FUSE) binding protein 3
*Lbx1 **	chr19:45232619-45235377	7.18	3.19	−1.17	2.05 × 10^−2^	ladybird homeobox homolog 1 (Drosophila)
*Maf **	chr8:115682935-115707940	9.66	5.98	−0.69	3.58 × 10^−2^	avian musculoaponeurotic fibrosarcoma oncogene homolog
*Nkx6-1 **	chr5:101658055-101665361	5.01	2.79	−0.85	3.58 × 10^−2^	NK6 homeobox 1
*Parp1*	chr1:180568937-180601389	12.93	102.04	2.98	2.05 × 10^−2^	poly (ADP-ribose) polymerase family, member 1
*Tcf24*	chr1:9960162-9967485	0.39	0.89	1.19	4.87 × 10^−2^	transcription factor 24
*Tcf7l2 **	chr19:55741714-55933693	6.56	3.44	−0.93	2.05 × 10^−2^	transcription factor 7 like 2, T cell specific, HMG box
**Losers vs. Control Mice**						
**Gene Symbol**	**Locus**	**Control Mice**	**Losers**	**log2 (Fold Change) ** **Losers/Controls**	**q_Value**	**Definition**
*Ctdspl*	chr9:118921134-119044353	5.14	26.56	2.37	0.0063	CTD (carboxy-terminal domain, RNA polymerase II, polypeptide A) small phosphatase-like
*E2f1 **	chr2:154559399-154569892	4.75	2.38	−0.99	0.0063	E2F transcription factor 1
*Egr1 ** ^,#,∆^	chr18:34861206-34864956	9.62	15.75	0.71	0.0115	early growth response 1
*En1 ** ^,#,§^	chr1:120602486-120607991	5.32	10.27	0.95	0.0063	engrailed 1
*En2 ** ^,#,∆^	chr5:28165695-28172166	3.09	5.95	0.94	0.0063	engrailed 2
*Fev ** ^,#,∆,¶,§^	chr1:74881508-74885408	1.25	3.95	1.66	0.0063	FEV (ETS oncogene family)
*Foxp2 ** ^,#^	chr6:14901169-15442450	1.59	3.15	0.98	0.0063	forkhead box P2
*Hoxb2*	chr11:96323125-96354174	3.72	0.64	−2.54	0.0063	homeobox B2
*Hoxb3*	chr11:96323125-96354174	2.98	0.31	−3.26	0.0063	homeobox B3
*Hoxd3*	chr2:74710043-74765142	1.82	0.08	−4.53	0.0115	homeobox D3
*Isl1 **	chr13:116298269-116309693	1.61	0.49	−1.72	0.0063	ISL1 transcription factor, LIM/homeodomain
*Lbx1 **	chr19:45232619-45235377	7.04	0.71	−3.32	0.0063	ladybird homeobox homolog 1 (Drosophila)
*Maf **	chr8:115682935-115707940	9.48	5.55	−0.77	0.0063	avian musculoaponeurotic fibrosarcoma oncogene homolog
*Mafb **	chr2:160363676-160367065	4.47	2.51	−0.84	0.0063	v-maf musculoaponeurotic fibrosarcoma oncogene family, protein B (avian)
*Neurod2 **	chr11:98325416-98329645	2.92	1.59	−0.88	0.0243	neurogenic differentiation 2
*Nkx6-1 **	chr5:101658055-101665361	4.91	1.66	−1.56	0.0063	NK6 homeobox 1
*Otx1*	chr11:21994763-22001651	0.62	2.50	2.01	0.0063	orthodenticle homeobox 1
*Otx2 **	chr14:48657676-48667644	0.73	2.03	1.49	0.01154	orthodenticle homeobox 2
*Pax2*	chr19:44746709-44838266	8.34	3.02	−1.47	0.0063	paired box 2
*Pax7 **	chr4:139737093-139832968	0.45	1.64	1.85	0.0063	paired box 7
*Phox2b **	chr5:67094327-67099126	7.03	1.56	−2.17	0.0063	paired-like homeobox 2b
*Rorc*	chr3:94372793-94398274	2.97	1.40	−1.08	0.0063	RAR-related orphan receptor gamma
*Scrt2*	chr2:152081528-152095802	13.30	7.04	−0.92	0.0063	scratch family zinc finger 2
*Sox14*	chr9:99874105-99876170	0.74	2.21	1.57	0.0063	SRY (sex determining region Y)-box 14
*Tal1*	chr4:115056425-115071758	2.45	4.25	0.80	0.0205	T cell acute lymphocytic leukemia 1
*Tbx20*	chr9:24720811-24774303	0.86	0.25	−1.77	0.0063	T-box 20
*Vdr ** ^,#^	chr15:97854426-97908296	0.76	0.29	−1.41	0.0115	vitamin D receptor
*Zkscan16*	chr4:58943559-58962705	7.72	12.67	0.71	0.0063	zinc finger with KRAB and SCAN domains 16

Genes associated with: *—behavior/neurological phenotype; ^#^—abnormal emotion/affect behavior; ^∆^—abnormal aggression-related behavior; ^¶^—increased aggression towards mice; **^§^**—abnormal fear/anxiety-related behavior.

**Table 3 ijms-21-06599-t003:** Metabolic pathways most significantly altered in defeated mice compared to controls.

Term	Count	*p* Value	Genes
Nicotine addiction	5	4.25 × 10^−4^	*Slc17a8*, *Gabrg2*, *Gabra1*, *Gabrb2*, *Chrna6*
Axon guidance	6	5.86 × 10^−3^	*Plxnc1*, *Nrp1*, *Sema7a*, *Sema3f*, *Ntng2*, *Slit1*
GABAergic synapse	5	7.55 × 10^−3^	*Gabrg2*, *Gad2*, *Gabra1*, *Gabrb2*, *Gad1*
Amyotrophic lateral sclerosis (ALS)	4	1.06 × 10^−2^	*Prph*, *Nefh*, *Nefl*, *Nefm*
Amphetamine addiction	4	2.20 × 10^−2^	*Ddc*, *Slc6a3*, *Th*, *Camk2a*
Chronic myeloid leukemia	4	2.65 × 10^−2^	*E2f1*, *Cdkn1a*, *Tgfb3*, *Mecom*
Bladder cancer	3	5.20 × 10^−2^	*E2f1*, *Cdkn1a*, *Mmp9*
Transcriptional misregulation in cancer	5	6.03 × 10^−2^	*Maf*, *Cdkn1a*, *Mmp9*, *Ngfr*, *Fev*
Retrograde endocannabinoid signaling	4	6.48 × 10^−2^	*Slc17a8*, *Gabrg2*, *Gabra1*, *Gabrb2*
Cocaine addiction	3	7.12 × 10^−2^	*Ddc*, *Slc6a3*, *Th*
Taurine and hypotaurine metabolism	2	9.44 × 10^−2^	*Gad2*, *Gad1*

**Table 4 ijms-21-06599-t004:** DEGs with changed expression both for the winners and losers (common DEGs).

Gene Symbol	Log2 (Fold_Change) Winners/Controls	q_Value	Log2 (Fold_Change) Losers/Controls	q_Value	Definition
*Apba3*	1.14	2.05 × 10^−2^	1.38	6.29 × 10^−3^	amyloid beta (A4) precursor protein-binding, family A, member 3
*Brd3*	−1.91	2.05 × 10^−2^	−1.69	6.29 × 10^−3^	bromodomain containing 3
*Cdk12*	−5.45	2.05 × 10^−2^	−5.43	6.29 × 10^−3^	cyclin-dependent kinase 12
*Dnaja1*	1.32	2.05 × 10^−2^	1.92	6.29 × 10^−3^	DnaJ heat shock protein family (Hsp40) member A1
*Egfl7*	−0.95	2.05 × 10^−2^	−0.97	6.29 × 10^−3^	EGF-like domain 7
*Gcn1l1*	−4.17	2.05 × 10^−2^	−3.96	6.29 × 10^−3^	GCN1 general control of amino-acid synthesis 1-like 1 (yeast)
*Gfap*	−0.90	2.05 × 10^−2^	−1.43	6.29 × 10^−3^	glial fibrillary acidic protein
*Gm13889*	−0.73	3.58 × 10^−2^	−0.82	6.29 × 10^−3^	predicted gene 13889
*Lbx1**	−1.17	2.05 × 10^−2^	−3.32	6.29 × 10^−3^	ladybird homeobox homolog 1 (Drosophila)
*Maf **	−0.69	3.58 × 10^−2^	−0.77	6.29 × 10^−3^	avian musculoaponeurotic fibrosarcoma oncogene homolog
*Myh14*	−0.64	4.87 × 10^−2^	−0.79	6.29 × 10^−3^	myosin, heavy polypeptide 14
*Myh6*	5.35	2.05 × 10^−2^	3.86	2.43 × 10^−2^	myosin, heavy polypeptide 6, cardiac muscle, alpha
*Myoc*	−1.89	2.05 × 10^−2^	−2.08	6.29 × 10^−3^	myocilin
*Ngfr*	−0.82	2.05 × 10^−2^	−0.86	1.59 × 10^−2^	nerve growth factor receptor (TNFR superfamily, member 16)
*Nkx6-1 **	−0.85	3.58 × 10^−2^	−1.56	6.29 × 10^−3^	NK6 homeobox 1
*Nrgn*	−0.66	3.58 × 10^−2^	0.96	6.29 × 10^−3^	neurogranin
*Prph*	−0.84	2.05 × 10^−2^	−2.22	6.29 × 10^−3^	peripherin
*Rln3*	−1.70	2.05 × 10^−2^	−3.54	6.29 × 10^−3^	relaxin 3
*Slc6a4*	0.99	2.05 × 10^−2^	2.20	6.29 × 10^−3^	solute carrier family 6 (neurotransmitter transporter, serotonin), member 4
*Smo*	−0.85	2.05 × 10^−2^	−0.82	3.74 × 10^−2^	smoothened, frizzled class receptor
*Taf1d*	4.61	2.05 × 10^−2^	4.58	6.29 × 10^−3^	TATA-box binding protein associated factor, RNA polymerase I, D
*Tph2*	0.87	4.87 × 10^−2^	1.87	6.29 × 10^−3^	tryptophan hydroxylase 2
*Vezt*	1.73	4.87 × 10^−2^	2.10	6.29 × 10^−3^	vezatin, adherens junctions transmembrane protein

* Genes, encoding transcription factors.

**Table 5 ijms-21-06599-t005:** Metabolic pathways most significantly altered in winners compared to defeated mice.

Gene Symbol	Mice		Log2 (Fold_Change) Losers/Winners	q_Value	Definition
	Winners FPKM	Losers FPKM			
**Amphetamine addiction, *p* Value = 3.30 × 10^−3^**					
*Camk2a*	22.69	33.70	0.57	6.12 × 10^−3^	calcium/calmodulin-dependent protein kinase II alpha
*Ddc*	21.21	36.21	0.77	6.12 × 10^−3^	dopa decarboxylase
*Fos*	3.97	6.68	0.75	6.12 × 10^−3^	FBJ osteosarcoma oncogene
*Slc6a3*	1.53	9.77	2.68	6.12 × 10^−3^	solute carrier family 6 (neurotransmitter transporter, dopamine), member 3
*Th*	12.78	22.60	0.82	6.12 × 10^−3^	tyrosine hydroxylase
**Dopaminergic synapse, *p* Value = 7.74 × 10^−3^**					
*Caly*	79.62	118.76	0.58	6.12 × 10^−3^	calcyon neuron-specific vesicular protein
*Camk2a*	22.69	33.70	0.57	6.12 × 10^−3^	calcium/calmodulin-dependent protein kinase II alpha
*Ddc*	21.21	36.21	0.77	6.12 × 10^−3^	dopa decarboxylase
*Fos*	3.97	6.68	0.75	6.12 × 10^−3^	FBJ osteosarcoma oncogene
*Slc6a3*	1.53	9.77	2.68	6.12 × 10^−3^	solute carrier family 6 (neurotransmitter transporter, dopamine), member 3
*Th*	12.78	22.60	0.82	6.12 × 10^−3^	tyrosine hydroxylase
**Adrenergic signaling in cardiomyocytes, *p* Value = 9.81 × 10^−3^**					
*Cacna2d3*	14.22	9.93	−0.52	2.68 × 10^−2^	calcium channel, voltage-dependent, alpha2/delta subunit 3
*Cacnb3*	14.49	21.10	0.54	6.12 × 10^−3^	calcium channel, voltage-dependent, beta 3 subunit
*Camk2a*	22.69	33.70	0.57	6.12 × 10^−3^	calcium/calmodulin-dependent protein kinase II alpha
*Scn4b*	73.35	50.16	−0.55	1.54 × 10^−2^	sodium channel, type IV, beta
*Scn5a*	0.43	0.84	0.99	6.12 × 10^−3^	sodium channel, voltage-gated, type V, alpha
*Tnnt2*	2.35	4.39	0.90	1.09 × 10^−2^	troponin T2, cardiac
**Amyotrophic lateral sclerosis, *p* Value = 1.14 × 10^−2^**					
*Nefh*	253.37	123.42	−1.04	6.12 × 10^−3^	neurofilament, heavy polypeptide
*Nefl*	710.19	436.14	−0.70	6.12 × 10^−3^	neurofilament, light polypeptide
*Nefm*	557.04	277.66	−1.00	6.12 × 10^−3^	neurofilament, medium polypeptide
*Prph*	26.44	10.15	−1.38	6.12 × 10^−3^	peripherin
**Axon guidance, *p* Value = 3.10 × 10^−2^**					
*Nrp1*	5.38	41.27	2.94	6.12 × 10^−3^	neuropilin 1
*Plxnc1*	3.03	4.52	0.58	1.54 × 10^−2^	plexin C1
*Sema3f*	5.26	8.44	0.68	4.25 × 10^−2^	sema domain, immunoglobulin domain (Ig), short basic domain, secreted, (semaphorin) 3F
*Slit1*	8.89	13.38	0.59	6.12 × 10^−3^	slit homolog 1 (Drosophila)
*Unc5d*	2.72	3.78	0.48	4.25 × 10^−2^	unc-5 netrin receptor D
**ECM-receptor interaction, *p* Value = 4.74 × 10^−2^**					
*Col27a1*	3.92	2.29	−0.78	6.12 × 10^−3^	collagen, type XXVII, alpha 1
*Spp1*	59.64	24.33	−1.29	6.12 × 10^−3^	secreted phosphoprotein 1
*Sv2b*	20.72	14.58	−0.51	1.09 × 10^−2^	synaptic vesicle glycoprotein 2 b
*Sv2c*	36.92	26.11	−0.50	1.09 × 10^−2^	synaptic vesicle glycoprotein 2c
**Tyrosine metabolism, *p* Value = 5.00 × 10^−2^**					
*Dbh*	21.59	11.27	−0.94	6.12 × 10^−3^	dopamine beta hydroxylase
*Ddc*	21.21	36.21	0.77	6.12 × 10^−3^	dopa decarboxylase
*Th*	12.78	22.60	0.82	6.12 × 10^−3^	tyrosine hydroxylase
**Cocaine addiction, *p* Value = 7.49 × 10^−2^**					
*Ddc*	21.21	36.21	0.77	6.12 × 10^−3^	dopa decarboxylase
*Slc6a3*	1.53	9.77	2.68	6.12 × 10^−3^	solute carrier family 6 (neurotransmitter transporter, dopamine), member 3
*Th*	12.78	22.60	0.82	6.12 × 10^−3^	tyrosine hydroxylase
**MAPK signaling pathway, *p* Value = 8.15 × 10^−2^**					
*Cacna1h*	3.63	5.58	0.62	6.12 × 10^−3^	calcium channel, voltage-dependent, T type, alpha 1H subunit
*Cacna2d3*	14.22	9.93	−0.52	2.68 × 10^−2^	calcium channel, voltage-dependent, alpha2/delta subunit 3
*Cacnb3*	14.49	21.10	0.54	6.12 × 10^−3^	calcium channel, voltage-dependent, beta 3 subunit
*Fos*	3.97	6.68	0.75	6.12 × 10^−3^	FBJ osteosarcoma oncogene
*Hspb1*	16.73	9.18	−0.87	6.12 × 10^−3^	heat shock protein 1
*Mecom*	1.06	0.62	−0.79	2.32 × 10^−2^	MDS1 and EVI1 complex locus

**Table 6 ijms-21-06599-t006:** Expression of genes that changed the level of transcription in comparisons of Losers vs. Control and Winners vs. Control in different directions and characterized as DEGs when comparing Losers vs. Winners.

Gene Symbol	Losers/Control				Winners/Control				Losers/Winners		Definition
	Controls FPKM	Losers FPKM	Log2 (Fold_Change) Losers/Controls	q Value	Controls FPKM	Winners FPKM	Log2 (Fold_Change) Winners/Controls	q Value	Log2 (Fold_Change) Losers/winners	q Value	
*Ercc2 ** ^,#,∆^	52.51	143.52	1.45	5.07 × 10^−2^	53.42	8.72	−2.62	2.05 × 10^−2^	4.06	6.12 × 10^−3^	excision repair cross-complementing rodent repair deficiency, complementation group 2
*Loxl2*	1.04	0.69	−0.59	9.99 × 10^−1^	1.06	46.25	5.45	2.05 × 10^−2^	−6.04	6.12 × 10^−3^	lysyl oxidase-like 2
*Nrgn* ^#^	21.95	42.78	0.96	6.29 × 10^−3^	22.35	14.15	−0.66	3.58 × 10^−2^	1.62	6.12 × 10^−3^	neurogranin
*Otx2 ** ^,#^	0.73	2.03	1.49	1.15 × 10^−2^	0.74	0.32	−1.19	6.96 × 10^−1^	2.67	6.12 × 10^−3^	orthodenticle homeobox 2
*Six3 **	1.93	6.56	1.77	7.09 × 10^−1^	1.96	0.68	−1.52	9.99 × 10^−1^	3.28	6.12 × 10^−3^	sine oculis-related homeobox 3
*Tcf7l2 ** ^,#^	6.44	9.72	0.59	1.07 × 10^−1^	6.56	3.44	−0.93	2.05 × 10^−2^	1.52	6.12 × 10^−3^	transcription factor 7 like 2, T cell specific, HMG box

***—DEGs encoding transcription factors; Genes associated with: ^#—^behavior/neurological phenotype; ^∆^—abnormal emotion/affect behavior.

**Table 7 ijms-21-06599-t007:** Metabolic pathways that include genes that maximize intergroup differences when comparing winners with defeated mice.

Term ID	Pathway	Count	*p* Value	Genes
mmu04916	Melanogenesis	8	8.60 × 10^−3^	*Wnt3*, *Adcy9*, *Adcy6*, *Mitf*, *Edn1*, *Crebbp*, *Camk2a **, *Fzd7*
mmu04020	Calcium signaling pathway	11	9.76 × 10^−3^	*Gnal*, *P2rx6*, *Slc8a2 **, *Adcy9*, *Ryr1*, *Cacna1g*, *Ppp3cc*, *Plcd4*, *Camk2a**, *Htr5a*, *Cacna1b*
mmu04713	Circadian entrainment	7	2.77 × 10^−2^	*Adcy9*, *Adcy6*, *Per2*, *Ryr1*, *Cacna1g*, *Per3*, *Camk2a **
mmu04141	Protein processing in endoplasmic reticulum	9	4.39 × 10^−2^	*Hsph1*, *Eif2ak1*, *Dnajb11*, *Man1a*, *Ubqln2*, *Ubqln1*, *Eif2ak2*, *Hspa8*, *Bcap31*
mmu04350	TGF-beta signaling pathway	6	4.95 × 10^−2^	*Acvr2b*, *Nog*, *Zfyve9*, *Crebbp*, *Chrd*, *Pitx2 **
mmu04261	Adrenergic signaling in cardiomyocytes	8	5.00 × 10^−2^	*Adcy9*, *Adcy6*, *Scn4b **, *Cacnb3 **, *Myh6*, *Cacna2d3 **, *Scn5a **, *Camk2a **

*—Genes, differentially expressed in VTA of winners and defeated mice (according to the Cufflinks/Cuffdiff programs).

**Table 8 ijms-21-06599-t008:** Correlation between expression of the key genes (*Nrgn*, *Camk2a*, *Ercc2*, *Otx2*, and *Six3)* and genes related to the dopamine signaling pathway.

**Correlation in VTA of Winners**					
**Gene Symbol**	***Nrgn***	***Camk2a***	***Ercc2***	***Otx2***	***Six3***
*Drd2*	0.722	0.920 **	0.605	0.895 *	0.902 *
*Slc6a3*	0.585	0.881 *	0.518	0.885 *	0.893 *
*Th*	0.517	0.696	0.202	0.527	0.537
*Ddc*	0.475	0.865 *	0.324	0.731	0.749
**Correlation in VTA of Losers**					
**Gene Symbol**	***Nrgn***	***Camk2a***	***Ercc2***	***Otx2***	***Six3***
*Drd2*	0.991 ***	0.970 **	0.909 *	0.924 **	0.944 **
*Slc6a3*	0.953 **	0.965 **	0.881 *	0.917 **	0.941 **
*Th*	0.910 *	0.789	0.747	0.651	0.703
*Ddc*	0.865 *	0.942 **	0.860 *	0.951 **	0.954 **

Level of significance (two-tailed test): * <0.05; ** <0.01; *** <0.001.

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
