# Peer review of "Gene Expression Changes in the Ventral Tegmental Area of Male Mice with Alternative Social Behavior Experience in Chronic Agonistic Interactions"

_ijms, 2020, doi:10.3390/ijms21186599_

Round 1

Reviewer 1 Report

The authors combine RNAseq analyses of the VTA to profile “winners” and “losers” identified using a social interaction test. The manuscript is well-written, with appropriate controls. The findings indicate a number of differentially regulated genes in winners and losers vs control as well as winners vs losers. Of interest to the authors were changes in the Nrgn gene and its involvement in dopaminergic signaling. The major findings presented serve to identify additional targets that would need more extensive future studies to link the changes in gene expression to a functional impact on VTA dopaminergic neurons and role in social stress-related behaviors. The following are provided to strengthen the manuscript.

  1. The behavioral data and separation of the phenotypes should be provided in light of transparency, rigor and reproducibility. Along these lines, the authors should justify the use of n=3 mice per group. Is this sufficiently powered for the analyses performed and interpretations made?
  2. It is important that the authors follow-up and validate the changes from the RNAseq data (especially for the Nrgrn and other highlighted genes) using an alternative method, e.g. qRT-PCR.
  3. In the Discussion, it would be beneficial for the authors to comment on the GABA system alongside the DA focus as the VTA is composed of both GABA and DA neurons.
  4. The readability of the tables could be improved by ensuring that all words and numbers are on the same line. These are large datasets and at times difficult for the reader to appreciate. Further, Table 7 is cut off in the manuscript.

Author Response

  1. The behavioral data and separation of the phenotypes should be provided in light of transparency, rigor and reproducibility. Along these lines, the authors should justify the use of n=3 mice per group. Is this sufficiently powered for the analyses performed and interpretations made?

Answer:  In our behavioral study we always used about as a rule more 10 animals per group to receive alternative types of social behaviors – winners or losers. At the end we separate the representative animals with most expressed behavioral phenotypes using our observations during all training time. RNA-seq method is very expensive, and do not allow to use large number of animals for analysis.    

Of course, more samples are always desirable for reliable results. But, firstly, statistics allows using n = 3. Second, we align animals very carefully for many behavioral metrics to avoid large scatter in the data. Then, even with n = 3, the statistics catch significant differences. An increase in the number of samples will only allow to reveal more DEGs. We would also like to note that researchers often use even two biological replicates in experiments related to the global measurements of gene expression.  According to the “Standards, Guidelines and Best Practices for RNA-Seq” from the ENCODE Consortium (http://genome.ucsc.edu/ENCODE/protocols/dataStandards/ENCODE_RNAseq_Standards_V1.0.pdf), experiments should be performed with two or more biological replicates.

So, the RNA-Seq studies published by other groups are often performed with n=2 or n=3:

·       Nagalakshmi U, Wang Z, Waern K, Shou C, Raha D, Gerstein M, Snyder M. The transcriptional landscape of the yeast genome defined by RNA sequencing. Science. 2008 Jun 6;320(5881):1344-9. doi: 10.1126/science.1158441. Epub 2008 May 1. PubMed PMID: 18451266; PubMed Central PMCID: PMC2951732. - “Two technical and two biological replicates were performed for each sample”.

  • Zhao S, Fung-Leung WP, Bittner A, Ngo K, Liu X. Comparison of RNA-Seq and microarray in transcriptome profiling of activated T cells. PLoS One. 2014 Jan 16;9(1):e78644. doi: 10.1371/journal.pone.0078644. eCollection 2014 . - “There were a total of six time points, with two biological replicates per time point.”
  • Dias C, Feng J, Sun H, Shao NY, Mazei-Robison MS, Damez-Werno D, Scobie K, Bagot R, LaBonté B, Ribeiro E, Liu X, Kennedy P, Vialou V, Ferguson D, Peña C, Calipari ES, Koo JW, Mouzon E, Ghose S, Tamminga C, Neve R, Shen L, Nestler EJ. β-catenin mediates behavioral resilience through Dicer1/microRNA regulation. Nature. 2014 December 4; 516(7529): 51–55. doi:10.1038/nature13976. (n=3 was used)
  • O Berton, CA. McClung, RJ. DiLeone, V Krishnan, WRenthal, SJ Russo, D Graham, NM. Tsankova, CA Bolanos, M Rios, LM Monteggia, DW Self, EJ. Nestler Essential Role of BDNF in the Mesolimbic Dopamine Pathway in Social Defeat Stress. 2006 VOL 311 SCIENCE www.sciencemag.org (n=2 and 3 were used)

In our experiment the statistical power of the intergroup differences with n=3 in each group (Fig. 1a) is characterized by t-value = 4.69; df=4; p=0.00937.

  1. It is important that the authors follow-up and validate the changes from the RNAseq data (especially for the Nrgn and other highlighted genes) using an alternative method, e.g. qRT-PCR.

Answer: Yes, you are correct, it is standard practice to use qRT-PCR to validate the quality of sequencing performed. In our case, on the contrary, we started with qRT-PCR for putative candidate genes (Th, Slc6a3, Bdnf, Snca) {Bondar, 2009 # 59} {Kudryavtseva, 2010 # 132}, and RNA-Seq analysis was performed later. Thus, several genes were previously tested in comparisons between experimental and control animals by qRT-PCR. Since the qRT-PCR data were published earlier, therefore, they are not repeated in this article, but only cited in the discussion. RNA-Seq is known to provide a much more accurate measurement of transcript levels than other methods (Wang Z, et al., Nat Rev Genet. 2009; 10: 57-63). In addition, several dedicated studies have shown that the results of the RNA-Seq show high levels of reproducibility, for both technical and biological replicates, and that the qRT-PCR and RNA-Seq results show a strong correlation (r=0.98) (Nagalakshmi U, et al., Science. 2008. 320: 1344-1349; Cloonan N, et al., Nat Methods. 2008. 5: 613-619).

Similar results and a correlation of 0.98 when comparing qRT-PCR and RNA-Seq was shown earlier and in several of our previous studies when profiling the level of gene transcription of a number of rat organs/tissues, including brain structures (Fedoseeva L.A., et al. BMC Genomics. 2019. 20(Suppl. 3):297; Fedoseeva L.A et al., BMC Genomics. 2016. 17(Suppl. 14):989; Ryazanova M.A et al., BMC Genetics. 2016. 17(Suppl. 3):151; Klimov L.O., et al. BMC Genetics. 2016. 17(Suppl. 1): 13).

In addition, we previously conducted extensive comparisons of qRT-PCR and RNA-Seq results in the midbrain raphe nuclei of male mice using >10 animals in in groups of winners and losers and found that the results were consistent between the methods (Boyarskikh et al., Mol Neurobiol. 2013 Aug;48(1):13-21; Smagin et al., Adv Biosci Biotech. 2013;4:10B 36-44; Kudryavtseva NN, et al., Mol Biol (Mosk). 2017;519(2):251-62). In order to cross-validate the obtained results, we employed the unique resource from Stanford University, USA (Zhang Y, et al., J Neurosci. 2014;34(36):11929-47) and found a significant concordance with our RNA-Seq data pool (Babenko VN, et al., J Integrat Bioinformat. 2017 Sep 13;14(3):20170024). These findings suggest that the transcriptome analyses of the data provided by the JSC Genoanalytica (http://genoanalytica.ru, Moscow) have been verified and that the method reflects the actual processes that occur in the brain structures taken into analysis.

However, we agree that sequencing and the comprehensive bioinformatics analysis carried out in this work, which made it possible to identify candidate genes for further analysis of their participation in the formation of behavioral features under the influence of social confrontations, is only the beginning of the process of studying the role of identified candidate genes (Nrgn and other highlighted genes), and additional experimental studies on proteome and metabolome levels, etc. are required. The corresponding statements have been added to the text of the manuscript (in the Limitations section) (Lines 453-458). 

  1. In the Discussion, it would be beneficial for the authors to comment on the GABA system alongside the DA focus as the VTA is composed of both GABA and DA neurons.

Answer: A discussion of changes in the level of transcription of genes related to the functioning of GABAergic synapses in both groups of experimental and control animals has been added to the Discussion section. See lines 290-302.

  1. The readability of the tables could be improved by ensuring that all words and numbers are on the same line. These are large datasets and at times difficult for the reader to appreciate. Further, Table 7 is cut off in the manuscript.

Answer: Tables are formatted.

The authors express their deep gratitude to the referee for carefully reading the manuscript and making comments, which made it possible to improve the presentation of the results of the work.

Reviewer 2 Report

Summary

The manuscript by Redina et al. compares using an RNA-Seq analysis the VTA transcriptome profiles of CD47 mice exposed to chronic social agonistic encounters. They differentiated between three different experimental groups depending on their social stress experiences: no-stress, chronically winning mice and losers. Animals within each group developed a different behavioral phenotype which was used as a selection criterion for the transcriptome analysis. With a final sample of 3 control mice, 3 winners and 3 losers their results showed that stress exerted a general effect inducing modifications in the levels of gene transcription when compared with non-stressed animals, being this effect greater on chronic losers than in winners. Finally, authors highlight that some genes (Nrgn, Ercc2, Otx2, and Six3) changed their VTA expression profiles in opposite directions in winners and in losers, positing that these genes may be key in the formation of the differential behavioral profile observed.

General comments

The major concern of the present work is that even if results are novel, authors did not highlight why are these results significant. I would like to clarify that I am not meaning that the results are not interesting or significant, what I considered is that authors should go further and explicit why is relevant to study the mechanism of formation of different behavioral patterns after social stress experiences. For instance, they could discuss if this study may be helping for a greater understanding of the genesis of some pathologic behaviors such as depression-like behavior or impulsive aggression. In this regard, I missed a clear reference of the translational value of the research.

I think that authors should highlight the significance of their research in the introduction. I also encourage them to include a conclusion section to briefly discuss the relevance and prospective of their study.

My second major concern is referred to the methodology. The experiment is apparently well designed and conducted, however is it necessary that authors include in the method section the procedure followed for the behavioral characterization of the different behavioral phenotypes. As behavior is used as a selection criterion for the transcriptome analysis, I consider that is crucial to address this issue in order to increase the replicability of the study.

Specific comments for each section

Abstract

After reading the abstract and the introduction the scope of the paper is not clear to me. While in the abstract it seems that the objective is to study VTA expression profiles in three different social stress conditions, not being this result related with any behavioral outcome. In the introduction (and also in the discussion), the scope of the paper is focused on the analysis differentially expressed VTA genes in winners and defeated animals (losers) compared with the controls as contributors in the formation of different aggressive/submissive behavior phenotypes. I strongly recommend that authors rewrite their abstract including the objective of linking the expression profile to the behavioral phenotype.

Introduction

While the information included in the Introduction section is enough for a good understanding of the study, I considered that the structure of this section could be improved. Three different sections of information can be distinguished: Lines 31-43 with information regarding the VTA; Lines 44-51 with information about the model of repeated agonistic interactions and its consequences; Lines 52-56 include previous results of changes in the VTA as a consequence of chronic agonistic stress. I think that a modification of the order of the narrative, by presenting first the model of social stress (Lines 44-51) followed by the previous results obtained in the VTA using the model (Lines 44-51) and ending with the information about the VTA (Lines 31-43) would enhance the understanding and readability of this part of the manuscript.

On the other hand, as mentioned previously in the “general comments” I think the quality of the introduction would increase if authors include a description of the relevance of their study.

Discussion

Line 379. Double period.

This section is mainly descriptive and helps for a better understanding of the results. However, I miss a final paragraph with a global view and significance of the results, limitation and prospective. As previously mentioned, I strongly recommend authors to include a conclusion section with this information.

Material and Methods

Line 402 and 441. It seems that there is a problem with the visualization of the “°” symbol.

4.1. Animals. I strongly recommend that authors include the initial sample size. The final samples are 3 control mice, 3 losers and 3 winners, however it is not specified how many animals were discarded due to their performance during the agonistic encounters, or any other reason.

It is stated that each experimental group is composed by a sample of 3 mice. Is seems an insufficient sample for the analysis carried out. Please justify sample size (n=3).

4.2. Generation of alternative forms of social behavior in male mice under agonistic interactions

Authors describe two different behavioral phenotypes (Lines 430-435) that were observed in winners and losers, and that were used as a selection criterion for the transcriptome analysis.

However, in the paper it is not explained the procedure followed for this behavioral characterization. I consider that is crucial to address this issue in order to increase the replicability of the study. Authors should explain which behaviors were evaluated and which instruments or protocols were employed.

Moreover, some characteristics (such as anxiety-like behavior) are shared between defeated and winner mice. I strongly recommend authors that clarify their selection criterion: which exact measures were used to characterize the behavioral profile and which standard or comparison was employed to determinate that animals displayed “pathological behaviors (Line 430)”. In this regard, I suggest that authors explain whether scores of each group (losers and winners) were compared to control group performance or were stablished considering pre- post- chronic defeat measures, or any other criteria.

During chronic social defeat stress protocols physical wounding of defeated mice is very common (Golden et al., 2011). Is it possible that the differential level of wounding between control mice, winners and losers could explain some of the obtained results? I think controlling for this variable and a discussion of this issue would increase the quality of the paper. I encourage authors to discuss this issue.

4.3. RNA-Seq analysis. Authors considered as bi-directionally changing the level of expression if this change is at least 1.5 times relative to the control in the control/winners and control/winners. Why is a change of at least 1.5 considered significant? I suggest that authors include a brief justification about why they pick this level.

Supplementary Materials

Line 492. Should read losers instead of loosers. This mistake is also present in the Supplementary material, Table 2 Tittle.

Author Response

The major concern of the present work is that even if results are novel, authors did not highlight why are these results significant. I would like to clarify that I am not meaning that the results are not interesting or significant, what I considered is that authors should go further and explicit why is relevant to study the mechanism of formation of different behavioral patterns after social stress experiences. For instance, they could discuss if this study may be helping for a greater understanding of the genesis of some pathologic behaviors such as depression-like behavior or impulsive aggression. In this regard, I missed a clear reference of the translational value of the research.

I think that authors should highlight the significance of their research in the introduction. I also encourage them to include a conclusion section to briefly discuss the relevance and prospective of their study.

Answer:  We followed your recommendation and added text in the introduction emphasizing the relevance of the work, and at the end of the manuscript we added a conclusion highlighting the importance and potential social significance of the results obtained.

My second major concern is referred to the methodology. The experiment is apparently well designed and conducted, however is it necessary that authors include in the method section the procedure followed for the behavioral characterization of the different behavioral phenotypes. As behavior is used as a selection criterion for the transcriptome analysis, I consider that is crucial to address this issue in order to increase the replicability of the study.

Answer:  Following specific comments detailed below, experimental details, test descriptions, and animal selection criteria for comparative profiling of gene transcription levels using RNA-Seq have been added to the Methods section.

Specific comments for each section

Abstract

After reading the abstract and the introduction the scope of the paper is not clear to me. While in the abstract it seems that the objective is to study VTA expression profiles in three different social stress conditions, not being this result related with any behavioral outcome. In the introduction (and also in the discussion), the scope of the paper is focused on the analysis differentially expressed VTA genes in winners and defeated animals (losers) compared with the controls as contributors in the formation of different aggressive/submissive behavior phenotypes. I strongly recommend that authors rewrite their abstract including the objective of linking the expression profile to the behavioral phenotype.

 Answer:  The abstract was revised according to your recommendations.

Introduction

While the information included in the Introduction section is enough for a good understanding of the study, I considered that the structure of this section could be improved. Three different sections of information can be distinguished: Lines 31-43 with information regarding the VTA; Lines 44-51 with information about the model of repeated agonistic interactions and its consequences; Lines 52-56 include previous results of changes in the VTA as a consequence of chronic agonistic stress. I think that a modification of the order of the narrative, by presenting first the model of social stress (Lines 44-51) followed by the previous results obtained in the VTA using the model (Lines 44-51) and ending with the information about the VTA (Lines 31-43) would enhance the understanding and readability of this part of the manuscript.

 Answer:  You are right, and we followed your recommendations.

On the other hand, as mentioned previously in the “general comments” I think the quality of the introduction would increase if authors include a description of the relevance of their study.

 Answer: The text describing the relevance of the study is included to the Introduction section. (Lines 71-75).

Discussion

Line 379. Double period.

Answer: Thank you. It was corrected.

This section is mainly descriptive and helps for a better understanding of the results. However, I miss a final paragraph with a global view and significance of the results, limitation and prospective. As previously mentioned, I strongly recommend authors to include a conclusion section with this information.

 Answer: A conclusion highlighting the importance of the findings and their potential social relevance, and a detailed description of the limitations of the study are added at the end of the Discussion section (see Lines  430-458).

Material and Methods

Line 402 and 441. It seems that there is a problem with the visualization of the “°” symbol.

Answer: Thanks a lot.  Corrected.

4.1. Animals. I strongly recommend that authors include the initial sample size. The final samples are 3 control mice, 3 losers and 3 winners, however it is not specified how many animals were discarded due to their performance during the agonistic encounters, or any other reason.

Answer: A group of 21 pairs of male C57BL/6J mice was used to generate alternative forms of social behavior.  This information has been added to the Methods section 4.2. The rest of the animals were not discarded, but used in various experiments.

It is stated that each experimental group is composed by a sample of 3 mice. Is seems an insufficient sample for the analysis carried out. Please justify sample size (n=3).

 Answer: Of course, more samples are always desirable for reliable results. But, firstly, statistics allows using n = 3. Second, we align animals very carefully for many behavioral metrics to avoid large scatter in the data. Then, even with n = 3, the statistics catch significant differences. An increase in the number of samples will only allow to reveal more DEGs.

We would also like to note that researchers often use even two biological replicates in experiments related to the global measurements of gene expression.  According to the “Standards, Guidelines and Best Practices for RNA-Seq” from the ENCODE Consortium (http://genome.ucsc.edu/ENCODE/protocols/dataStandards/ENCODE_RNAseq_Standards_V1.0.pdf), experiments should be performed with two or more biological replicates.

So, the RNA-Seq studies published by other groups are often performed with n=2 or n=3:

·       Nagalakshmi U, Wang Z, Waern K, Shou C, Raha D, Gerstein M, Snyder M. The transcriptional landscape of the yeast genome defined by RNA sequencing. Science. 2008 Jun  6; 320(5881):1344-9. doi: 10.1126/science.1158441. Epub 2008 May 1. PubMed PMID: 18451266; PubMed Central PMCID: PMC2951732. - “Two technical and two biological replicates were performed for each sample”

  • Zhao S, Fung-Leung WP, Bittner A, Ngo K, Liu X. Comparison of RNA-Seq and microarray in transcriptome profiling of activated T cells. PLoS One. 2014 Jan 16;9(1):e78644. doi: 10.1371/journal.pone.0078644. eCollection 2014 . - “There were a total of six time points, with two biological replicates per time point.”
  • Dias C, Feng J, Sun H, Shao NY, Mazei-Robison MS, Damez-Werno D, Scobie K, Bagot R, LaBonté B, Ribeiro E, Liu X, Kennedy P, Vialou V, Ferguson D, Peña C, Calipari ES, Koo JW, Mouzon E, Ghose S, Tamminga C, Neve R, Shen L, Nestler EJ. β-catenin mediates behavioral resilience through Dicer1/microRNA regulation. Nature. 2014 December 4; 516(7529): 51–55. doi:10.1038/nature13976. (n=3 was used)
  • O Berton, CA. McClung, RJ. DiLeone, V Krishnan, WRenthal, SJ Russo, D Graham, NM. Tsankova, CA Bolanos, M Rios, LM Monteggia, DW Self, EJ. Nestler Essential Role of BDNF in the Mesolimbic Dopamine Pathway in Social Defeat Stress. 2006 VOL 311 SCIENCE www.sciencemag.org (n=2 and 3 were used)

In our experiment the statistical power of the intergroup differences with n=3 in each group (Fig. 1a) is characterized by t-value = 4.69; df=4; p=0.00937.

4.2. Generation of alternative forms of social behavior in male mice under agonistic interactions

Authors describe two different behavioral phenotypes (Lines 430-435) that were observed in winners and losers, and that were used as a selection criterion for the transcriptome analysis.

However, in the paper it is not explained the procedure followed for this behavioral characterization. I consider that is crucial to address this issue in order to increase the replicability of the study. Authors should explain which behaviors were evaluated and which instruments or protocols were employed.

Answer: the following text was added to the Methods section 4.2:

The following criteria were used. For the losers, during activation period (5 min before a fight) the chronically defeated mice demonstrated all symptoms of depressive behavior: they did not approach the partition, sat in the corner of the cage opposite to the partition, or with their nose into a corner or into litter; they were characterised by immobility, freezing under winners’ attacking, demonstrated indifference in all experimental situation (without behavioral reactions), there were no inversions of behavior to the opposite one after changes of aggressors, showed avoidance and passive defense when attacked by the aggressor. Aggressive winners, during activation period demonstrated strong aggressive motivation, and every day immediately attacked the losers after partition removing, stopping only for rest, displaying manic motivation to bite losers in spite on full submission.  

Moreover, some characteristics (such as anxiety-like behavior) are shared between defeated and winner mice. I strongly recommend authors that clarify their selection criterion: which exact measures were used to characterize the behavioral profile and which standard or comparison was employed to determinate that animals displayed “pathological behaviors (Line 430)”. In this regard, I suggest that authors explain whether scores of each group (losers and winners) were compared to control group performance or were established considering pre- post- chronic defeat measures, or any other criteria.

Answer:  Both groups are characterized by anxiety and that is absolutely right.  We have already discussed this side of the issue in our earlier articles: Avgustinovich D.F., Gorbach O.V., Kudryavtseva N.N. (1997) Comparative analysis of anxiety-like behavior in partition and plus-maze tests after agonistic interactions in mice. Physiol. Behav. 61(1), 37-43.Kudryavtseva N.N., Bondar N.P., Avgustinovich D.F. (2002) Associationbetween experience of aggression and anxiety in male mice. Behav. BrainRes. 133(1), 83-93.Smagin D.A., Park J-H, Michurina T.V., Peunova N., Glass Z., Sayed K., Bondar N.P., Kovalenko I.L., Kudryavtseva N.N., Enikolopov G. (2015) Altered hippocampal neurogenesis and amygdalar neuronal activity in adult mice with repeated experience of aggression. Front. Neurosci.9:443. doi: 10.3389/fnins.2015.00443.Galyamina, A.G.; Kovalenko, I.L.; Smagin, D.A.; Kudryavtseva, N.N. Interaction of depression and anxiety in the development of mixed anxiety/depression disorder: Experimental studies of the mechanisms of comorbidity (review). Neurosci. Behav. Physiol. 2017, 47, 699–713.

The results presented in this manuscript are in good agreement with these concepts. We added several words clarifying this point into Discussion section (Lines 304-305).

During chronic social defeat stress protocols physical wounding of defeated mice is very common (Golden et al., 2011). Is it possible that the differential level of wounding between control mice, winners and losers could explain some of the obtained results? I think controlling for this variable and a discussion of this issue would increase the quality of the paper. I encourage authors to discuss this issue.

Answer:  Thank you. This point was clarified in Methods section 4.2., Lines 482-483.

4.3. RNA-Seq analysis. Authors considered as bi-directionally changing the level of expression if this change is at least 1.5 times relative to the control in the control/winners and control/losers. Why is a change of at least 1.5 considered significant? I suggest that authors include a brief justification about why they pick this level.

Answer:  We did not write that this level corresponds to statistical reliability. The statistical significance is assessed separately. This relationship is actually similar to effect size, i.e. emphasizes the difference that is clinically or biologically meaningful between differentially expressed genes. It is unknown in advance and its meaning is chosen rather arbitrarily on the basis of previous experience or from the literature. When studying gene expression, levels of 1.5 or 2 are most often chosen, as can be seen from the publications below:

  • “Transcripts or genes with FDR ≤ 0.05 and |Log2FC| ≥ 1 were considered as significant DEGs” (Cun Liu, et al., BMC Genomics. 2019; 20: 774).
  • “Differentially expressed genes for each platform were identified by comparison of the mean of expression intensities or counts of compound-treated samples to the mean of expression intensities or counts of the corresponding vehicle-treated samples with a fold change (FC) > 1.5 and p < 0.01” (Mohan S. Rao et al., Front Genet. 2018; 9: 636).
  • “Only genes that exhibited changes in expression > 1.5-fold and had a P-values adjusted using the Benjamini–Hochberg procedure lower 0.05 (Padj < 0.05) were considered” (V. Dergunova et al., BMC Genomics. 2018; 19: 655).
  • “DEGs were identified by using an unpaired Student’s test with a P-value cutoff of 0.05 and fold change greater than 2.0 (upregulation) or less than 0.5 (downregulation)” (Xun Li et al., Cell Physiol Biochem . 2018;51(5):2198-2211).
  • “The Cuffdiff 2 software was used to identify the DEGs between the estrus and proestrus groups using the following filter criteria: P value <0.05 and absolute value of log2 (FPKM_AGED/FPKM_YOUNG) >1” (Liang Zhang et al., Am J Transl Res. 2019; 11(10): 6553–6560).

Supplementary Materials

Line 492. Should read losers instead of loosers. This mistake is also present in the Supplementary material, Table 2 Tittle.

Answer: Thank you very much. Corrections made.

The authors express their deep gratitude to the referee for carefully reading the manuscript and making comments, which made it possible to improve the presentation of the results of the work.

Reviewer 3 Report

The manuscript “Ventral Tegmental Area profiling to discover mechanisms underlying the formation of alternative behavior patterns in male mice under daily social agonistic interactions” by Olga Redina, Vladimir Babenko, Dmitry Smagin, Irina Kovalenko, Anna Galyamina, Vadim Efimov, and Natalia Kudryavtseva is aimed at solving high-priority tasks directed to identifying candidate genes associated with the manifestation of aggressive and depressive behaviour under conditions of social stress.  

The paper presents the results of a comparative analysis of gene transcription in the VTA of male mice with an alternative type of behaviour, which was experimentally formed during daily animal confrontations. Differences in gene expression are shown both when comparing winners and losers with a group of control animals, and when comparing two experimental groups with each other.

A comprehensive bioinformatic analysis was carried out using modern methods and databases, which allowed, among a large number of differentially expressed genes, to identify several that can reasonably be considered the most likely candidate genes involved in the formation of alternative forms of behaviour during social confrontations, at least in those experimentally specified conditions that were used in the work.
There are no fundamental remarks on the performance of the work, analysis and presentation of the results; however, there are several minor comments that need to be corrected before publication.  

Minor comments:

Throughout the text, authors should use italics for gene symbols

Table 1.: according to the name of the Table (DEGs associated with the behaviour / neurological phenotype), all genes listed in it are known as associated with the behaviour / neurological phenotype. What was the point of marking them all without exception with an asterisk and signing that they are associated with the behaviour / neurological phenotype?

Table 1.: log2 (fold change) W/C - it is necessary to give decryption for W / C
Table 1.log2 (fold change) L/C - it is necessary to give decoding for L / C
Table 2.log2 (fold change) A/C - it is necessary to give decryption for A / C
Table 2.log2 (fold change) L/C - it is necessary to give decoding for L / C

Line 95: Among the DEG encoding transcription factors, 16 genes are associated with behaviour / neurological phenotype.

– please, use the plural form – DEGs

Table 6.    it is necessary to give decoding for L/C, W/C, L/W

Lines 165-166: It is not clear from the text which genes are included in Supplementary Table 6. More precisely, first, it is declared, and only then (in the next paragraph) it is explained which genes are included in it.

Line 333: There are only 65 references in the work, so how to understand the reference in the following sentence: “Given that Ercc2 is involved in repairing damage caused by the redox process [38 73], it can be assumed that oxidative stress processes in the VTA play a pivotal role in the formation of phenotypic features of defeated mice ”.  

Author Response

Minor comments:

Throughout the text, authors should use italics for gene symbols

Answer: Thank you for the comment. All gene symbols have been italicized.

Table 1.: according to the name of the Table (DEGs associated with the behaviour / neurological phenotype), all genes listed in it are known as associated with the behaviour / neurological phenotype. What was the point of marking them all without exception with an asterisk and signing that they are associated with the behaviour / neurological phenotype?

Answer: Thank you. Extra markings have been removed from the Table 1.

Table 1.: log2 (fold change) W/C - it is necessary to give decryption for W / C
Table 1.log2 (fold change) L/C - it is necessary to give decoding for L / C
Table 2.log2 (fold change) A/C - it is necessary to give decryption for A / C
Table 2.log2 (fold change) L/C - it is necessary to give decoding for L / C

Answer: Thank you. Corrections have been inserted to Tables 1 and 2.

Line 95: Among the DEG encoding transcription factors, 16 genes are associated with behaviour / neurological phenotype.

– please, use the plural form – DEGs

Answer: This and other corrections of English have been made.

Table 6.    it is necessary to give decoding for L/C, W/C, L/W

Answer: Corrections have been inserted to Table 6.

Lines 165-166: It is not clear from the text which genes are included in Supplementary Table 6. More precisely, first, it is declared, and only then (in the next paragraph) it is explained which genes are included in it.

Answer: The text was corrected.

Line 333: There are only 65 references in the work, so how to understand the reference in the following sentence: “Given that Ercc2 is involved in repairing damage caused by the redox process [38 73], it can be assumed that oxidative stress processes in the VTA play a pivotal role in the formation of phenotypic features of defeated mice ”.  

Answer: Thank you for the comment. Citation has been corrected.

The authors express their deep gratitude to the referee for carefully reading the manuscript and making comments, which made it possible to improve the presentation of the results of the work.

Round 2

Reviewer 1 Report

The authors were responsive to my previous comments and concerns.

Author Response

The authors are sincerely grateful to the reviewer for carefully reading the manuscript and all the comments and concerns expressed, which helped to significantly improve the article.